# Ecosystem health shapes viral ecology in peatland soils

**James C. Kosmopoulos** [1,2], **William Pallier**[3], **Ashish A. Malik** [3,4,7] & **Karthik Anantharaman** [1,5,6,7]

Peatlands hold up to one-third of Earth's soil carbon but are increasingly turning from being carbon sinks to becoming carbon sources due to human impacts. Restoration efforts aim to reverse this trend, but viral influences on peatland recovery remain unclear, despite viruses being potent regulators of microbiomes and ecosystem function. Here we sequenced soil metagenomes to study viral communities across seven UK peatlands, each encompassing areas representing three peatland ecosystem health statuses: natural, damaged and restored. We found that viral diversity and community structure were shaped by both geography and ecosystem health. Viruses were geographically widespread, yet exhibited ecosystem health-specific endemism and functional adaptation, highlighting their sensitivity to restoration. Virus–host dynamics ranged from stable 'piggyback-the-winner' relationships to decoupled dynamics in those infecting keystone aerobes, sulfate reducers, carbohydrate degraders and fermenters. These findings position viruses as dynamic drivers of peatland ecosystem recovery and could unlock pathways to bolster carbon retention and accelerate climate mitigation.

Peatlands are globally important ecosystems and the largest terrestrial carbon store[1–4]. Despite covering just 3% of the Earth's surface, they are estimated to contain up to one-third of the global soil carbon due to accumulation of undecomposed organic matter in anoxic, waterlogged soils[1–4]. However, peatlands are sensitive to environmental disturbances such as drainage and desiccation caused by land-use changes, climate change and other anthropogenic impacts[5–7]. When peatlands are degraded, they shift from being carbon sinks to becoming carbon sources, releasing stored carbon as $CO_2$ and exacerbating global climate change[7–9].

Restoration efforts, often focused on rewetting, aim to reverse these effects by restoring the natural hydrology that maintains peatland carbon storage, but outcomes for soil carbon storage and function are variable, highlighting a need to identify key ecological drivers of recovery[10–12]. Soil microbiomes are central to peatland function,

regulating carbon retention and loss[3,13]. Restoration success probably depends on reestablishing vital microbial processes alongside hydrology and vegetation. Despite their known influence on microbiomes and global biogeochemistry, the roles of viruses in peatland ecology and recovery remain poorly understood.

Soil viruses are now recognized as ubiquitous members of microbiomes and potent regulators of nutrient cycles, as shown in marine environments. Research on peatland viruses is nascent, but pioneering studies indicate that viruses can strongly influence carbon cycling. For instance, a peatland undergoing permafrost thaw revealed viral communities that respond strongly to environmental change: as permanently frozen 'bogs' transition to wetter, thawed 'fens', viral community composition shifts from 'soil-like' viruses to 'aquatic-like' assemblages, mirroring changes in hydrology and host communities[14]. In some peatlands, viruses encode auxiliary metabolic genes (AMGs) that may enhance

[1]Department of Bacteriology, University of Wisconsin-Madison, Madison, WI, USA. [2]Microbiology Doctoral Training Program, University of Wisconsin-Madison, Madison, WI, USA. [3]School of Biological Sciences, University of Aberdeen, Aberdeen, UK. [4]School of GeoSciences, University of Edinburgh, Edinburgh, UK. [5]Department of Integrative Biology, University of Wisconsin-Madison, Madison, WI, USA. [6]Department of Data Science and AI, Wadhwani School of Data Science and AI, Indian Institute of Technology Madras, Chennai, Tamil Nadu, India. [7]These authors jointly supervised this work: Ashish A Malik, Karthik Anantharaman. ✉e-mail: ashish.malik@ed.ac.uk; karthik@bact.wisc.edu

carbon cycling[14–17]. These findings indicate that peatland soil viruses are not passive bystanders, but dynamic, active drivers of carbon flow and ecosystem function. However, comprehensive understanding of their interactions with microbiomes and peatland health remains lacking.

As peatlands are increasingly targeted for restoration to mitigate climate change and preserve biodiversity, understanding how viral dynamics influence microbial processes is crucial for predicting and improving restoration outcomes. We hypothesized that (1) peatlands of varying levels of restoration harbour unique viral assemblages; (2) viral populations mirror host shifts across ecosystem states, including in metabolic contexts; and (3) increased microbial activity in damaged sites favours 'piggyback-the-winner' dynamics, boosting lysogeny and favouring fast-growing hosts. By examining the relationships between viral communities, microbial hosts and environmental factors, our study enhances our understanding of the ecological and functional roles of viruses in peatlands.

## Results

### Viral communities in UK peatlands

We sampled soils from seven UK peatlands (Fig. 1a and Supplementary Table 1) spanning a gradient of ecosystem health statuses (EHSs): near-natural (undrained/undisturbed reference, hereafter natural), damaged (drained/eroded) and restored (formerly damaged, then rewetted)[18]. Damaged peatlands were probably drained for decades, although the exact duration is unknown. Restoration age varied by site but was within 10 years of sampling. Compared with natural sites, damaged sites have drier, more oxygenated soils and higher community-wide growth rates[18]. Restored sites show signs of recovery but remain chemically distinct from natural peatlands[18].

We sequenced community DNA of soil samples and co-assembled metagenomes by combining triplicate sequence read libraries from each sampling site and EHS, yielding 22 assemblies (one per site × EHS combination) of high quality (Supplementary Tables 1 and 2). We identified 3,177 viral scaffolds across all sites and EHS from metagenome co-assemblies, which were binned into 2,281 viral metagenome-assembled genomes (vMAGs) (Supplementary Table 3). These genomes were dereplicated and clustered into 1,548 virus species-level clusters which were analysed downstream using virus species-level representative genomes.

### Environmental differences between EHSs

A principal components analysis (PCA) across sites revealed that EHS generally reflected the composition of environmental parameters (Extended Data Fig. 1a). PCA loadings indicated that total carbon (0.72), pH (0.63) and oxygen concentration (0.61) were the strongest parameters, with moisture (0.57), total nitrogen (0.51) and conductivity (0.38) being also important, supporting EHS's role in structuring peatland soils. Site-specific effects were strong, as shown by separate PCAs within sites, where the relative influence of environmental variables differed (Extended Data Fig. 1b). To capture the complex variation across sites and EHSs, an ecosystem health index (EHI) was previously calculated for each sample[18] (Fig. 1a), incorporating peat chemistry, oxygen, moisture and vegetation. This index provides a holistic, continuous measure of peatland ecosystem health that effectively reflects variation across all samples.

While EHS grouped samples within sites, the degree of separation varied, suggesting that local environmental conditions matter. This is consistent with the fact that these sites span a climatic gradient and vary in their level of degradation and restoration[18]. Nevertheless, soils from damaged peatlands were less waterlogged, more oxygenated and more acidic compared with the natural contrast where soils from restored peatlands demonstrated signs of mitigation. Overall, these results indicate that while site-level differences are prominent, EHS captures meaningful environmental variation across peatland sites. Likewise, changes in EHI offer a useful means to track relative improvements in ecosystem health when comparing areas of varying EHSs within a given peatland.

### Geography and ecosystem health structure peatland virus communities

To explore the drivers of viral community composition, we performed principal coordinate analysis (PCoA) and found that geography was the primary structuring factor (Fig. 1b). Although the viral communities of some samples from different sites were similar in composition, samples in the PCoA were mostly grouped by their geographic origin ($R = 0.656$, $P = 0.001$, analysis of similarities (ANOSIM)). The influence of EHS on community structure became more apparent when analysing sites separately. Within sites, we observed a strong separation of samples by EHS ($R > 0.5$, $P < 0.05$, ANOSIM; Fig. 1c). The exception was Stean, where lack of natural reference samples probably reduced power. Alongside overall EHS grouping, we mapped the previously calculated EHI for each sample[18] to our PCoA (Fig. 1d) and found that EHI also significantly impacted viral community structure ($R^2 = 0.029$, $P = 4.5 \times 10^{-3}$, PerMANOVA; Supplementary Table 4) independently of EHS ($R^2 = 0.051$, $P = 1 \times 10^{-4}$). EHI was also strongly positively correlated with virus community PCo1 (Fig. 1e), providing further evidence that ecosystem health is a significant factor that drives viral community structure. Host community composition (Extended Data Fig. 2) also significantly impacted viral community structure, but this did not overshadow the independent effects of sample site and ecosystem health (Supplementary Table 4 and Supplementary Results).

### Peatland soils contain a mix of endemic and shared viral populations

Considering the strong effects of geography and ecosystem health on structuring environmental variation across sites, we examined the degree of endemism among our identified virus species. Most viral genomes were detected in soil metagenomes from multiple sample sites (76% of species representatives, Fig. 2a). However, 54% of virus species were endemic to individual EHS (found exclusively in one of natural, damaged or restored soils across all sites) compared with 46% that were shared (Fig. 2b). We also assessed whether the viral genomes identified were largely novel or instead represented in published soil virus databases. We gathered a comprehensive collection of genomes from the three largest and most recent soil virus databases[17,19,20]. We found that more viruses from this study formed genus-level genome clusters with viruses from other databases than other viruses from this study (Fig. 2c). Thus, many viral genomes clustered with known soil viral genomes from other ecosystems, indicating that not all are unique to peatlands at the genus level. These results suggest that soil viruses in UK peatlands share a core of virus lineages with other soils, alongside a substantial fraction of locally endemic viruses.

### Viruses are differentially abundant across EHSs

Having established that ecosystem health significantly shapes viral communities, we next identified viruses that were differentially abundant across EHSs. Using DESeq2 (ref. 21), we created ecosystem health 'trend groups' for a qualitative analysis of functions (Supplementary Results and Extended Data Fig. 3). Host genomes were also differentially abundant across EHSs and were clustered into trend groups (EHS group; Supplementary Results and Extended Data Fig. 4). For detailed information on the distribution and clustering of viral species-representative genomes, see Supplementary Results. Across all sites, there was a greater proportion of damaged-enriched viruses (37%) than restored-enriched viruses (33%) and natural-enriched viruses (29%) (Fig. 3a). This contrasted with trends for hosts, indicating that damaged peatlands host a greater share of enriched viruses among differentially abundant groups. In summary, the differential abundance of viral species across EHSs shows that environmental health strongly influences viral population sizes, which vary strongly between natural, restored and damaged peatland soils.

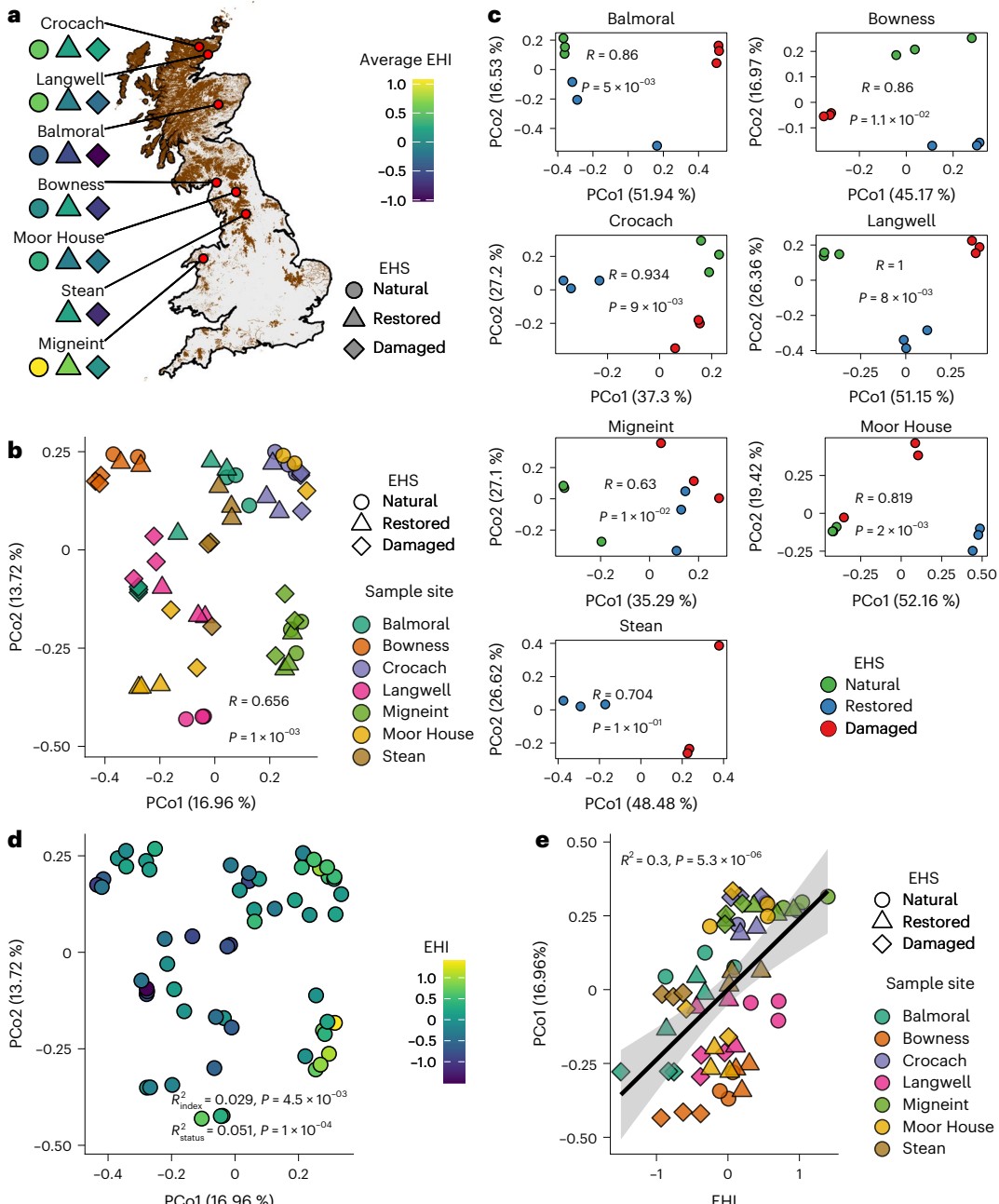

**Fig. 1 | Geography and ecosystem health structure peatland environmental variation. a**, Map of the seven peatland sites. Natural, restored and damaged sampling blocks of EHS are coloured by their average EHI. The map shading indicates peatland cover. No natural blocks were sampled at Stean. **b**, PCoA of viral community Bray–Curtis dissimilarities ($n = 60$ soil samples). ANOSIM (999 permutations) shows significant separation by site ($R = 0.656$, $P = 1.0 \times 10^{-3}$, unadjusted). **c**, Site-specific PCoAs with ANOSIM statistics for separation of communities by EHS (999 permutations; exact $R$ and $P$ values shown on plots).

**d**, The PCoA from **b** coloured by EHI. PerMANOVA (two-sided, 999 permutations) reports the marginal $R^2$ for variance in community dissimilarity explained by EHI ($R^2_{index} = 0.029$, $P = 4.5 \times 10^{-3}$ BH-adjusted) and EHS ($R^2_{status} = 0.051$, $P = 1.0 \times 10^{-4}$, BH-adjusted). **e**, Linear regression of PCoA axis 1 against EHI, where the black line represents the fitted regression mean and the shaded band indicates the 95% confidence interval around the fitted line. Regression statistics from a linear model of the two axes are provided ($R^2 = 0.30$, $P = 5.28 \times 10^{-6}$).

## The abundance of viruses across EHSs is discordant with dominant host taxa

We examined the relative abundance of differentially abundant bacterial and archaeal MAGs and their predicted viruses at the host class level within each EHS (Fig. 3b). Relative abundances of viruses and their hosts varied substantially across sites. However, when examining all sites together, viruses infecting hosts in the phyla Actinomycetota, Desulfobacterota and Planctomycetota showed a marked decrease in abundance from the natural to the restored trend groups. This was met with an increase in viruses infecting Pseudomonadota hosts, particularly Alphaproteobacteria. Viruses of Alphaproteobacteria and Desulfobaccia also increased in abundance from natural to damaged trend groups. These viral abundance shifts did not mirror host changes. For example, while the relative abundance of Pseudomonadota viruses surged from natural to restored groups, the abundance of Pseudomonadota hosts remained stable. Similarly, Desulfobaccia hosts made up only 1.8% of damaged-enriched hosts, yet Desulfobaccia viruses represented 6.6% of damaged-enriched viruses. These findings show

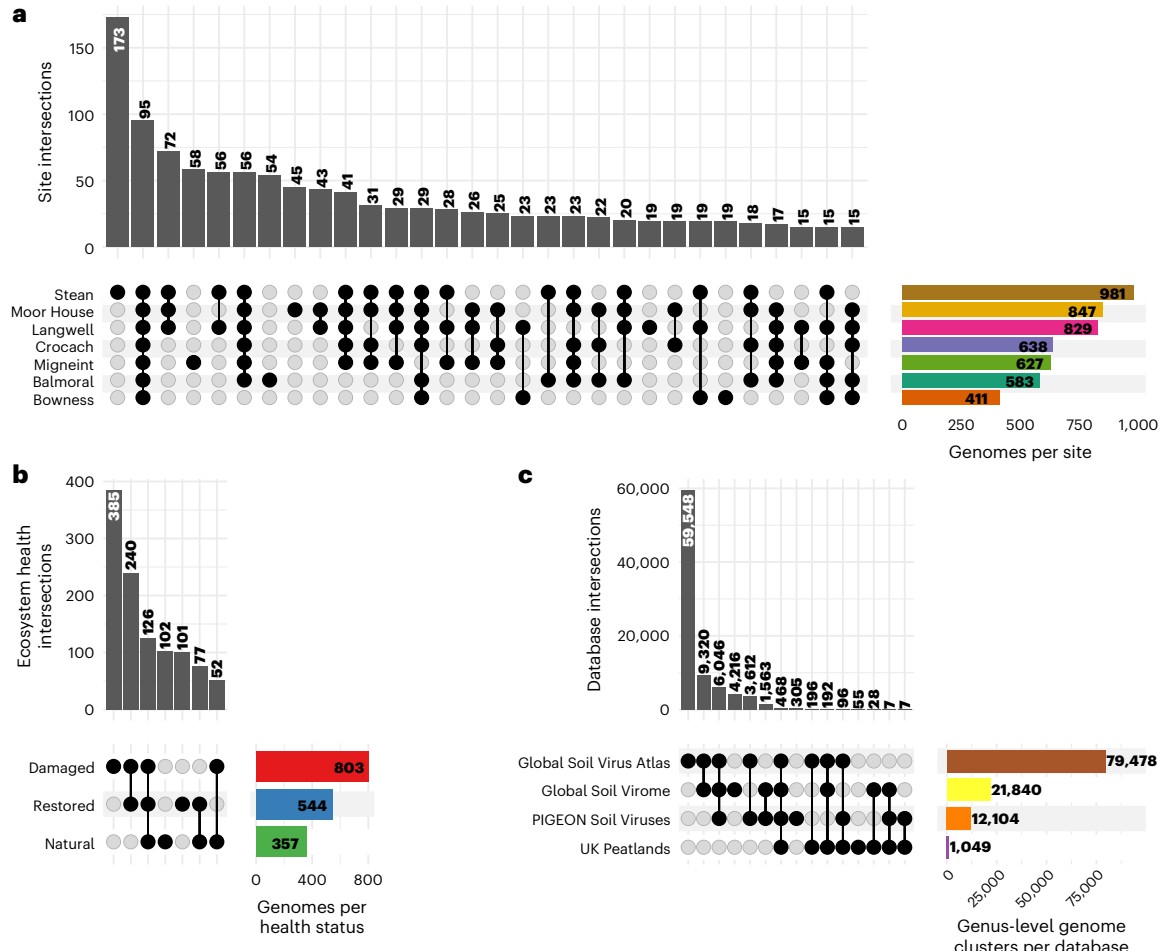

**Fig. 2 | UK peatland soil viruses are widely distributed across EHSs, sample sites and databases. a,b**, Detection of species-representative virus genomes across soils from different sample sites (**a**) and soil with different EHSs (**b**). For simplicity, intersections with <15 genomes are omitted in **a** and **b**. **c**, Intersections of databases represented in genus-level clusters of the viruses here and other genomes from three soil virus databases[17,19,20] (*n* = 729,998 genomes). Bars represent clusters with genomes originating from each database in the intersection. Numbers above or inside bars indicate the total number of genome clusters with genomes originating from each database in the intersection.

that viral and host dynamics across EHSs are discordant, suggesting that viral responses to environmental changes may depend on factors beyond host availability.

## Viral and host dynamics across key biogeochemical functions in peatlands

Given that peatlands at different EHSs are chemically distinct, we explored whether viruses infecting microbes with key biogeochemical functions changed across EHSs, and whether these changes reflected overall viral and host abundance trends. We calculated the relative abundance of viruses infecting hosts with eight metabolic functions (Supplementary Table 5) within each EHS group, normalized by overall viral abundance within that same trend group (Fig. 3c). Similar calculations were done for predicted hosts. Below, we focus on notable trends, but comprehensive results for all eight metabolic functions are provided in Supplementary Results.

Across the eight metabolic functions, several key trends stood out. For oxidative phosphorylation, viral abundance (*n* = 451 virus genomes) increased from natural (1.00) to restored soils (1.12, +11%), but decreased again from restored to damaged (0.89, −25%). Host abundance (*n* = 365 host genomes), in contrast, decreased from natural to restored (0.82, −32%). For fermentation, viral abundance (*n* = 491 virus genomes) remained stable between natural (1.07) and restored soils (1.08, +1.7%) but showed a decrease in damaged soils

(0.87, −22%), paralleling a similar decrease in host abundance (−32%, *n* = 477 host genomes). Carbohydrate degradation showed minor changes, with viral abundance (*n* = 542 virus genomes) being stable between natural and restored soils (1.05) but decreasing slightly in damaged soils (0.91, −15%), alongside minor fluctuations in host abundance (*n* = 590 host genomes). For assimilatory and dissimilatory sulfate reduction, there were decreases in viral abundance (−52% and −36%, *n* = 20 and *n* = 105 virus genomes) from natural to restored soils, accompanied by an even greater decrease for their hosts (−159% and −48%, *n* = 34 and *n* = 68 host genomes). From restored to damaged soils, viruses infecting hosts with these functions increased by 25% (assimilatory) and 17% (dissimilatory). Although assimilatory sulfate-reducing hosts showed a major increase of 70% from restored to damaged soils, dissimilatory sulfate-reducing hosts declined by 25% over the same transition, contrasting with the pattern observed for their viruses. A similar pattern was observed for thiosulfate oxidation (*n* = 33 virus genomes, *n* = 16 host genomes). It is important to note that these percentage changes reflect descriptive trends based on aggregated ratios and were not subjected to null hypothesis testing (see Methods). These patterns suggest that while virus and host dynamics often align, the enrichment of hosts with specific metabolic functions, such as oxidative phosphorylation and sulfur cycling, can sometimes diverge from the enrichment of viruses that infect them across different EHSs.

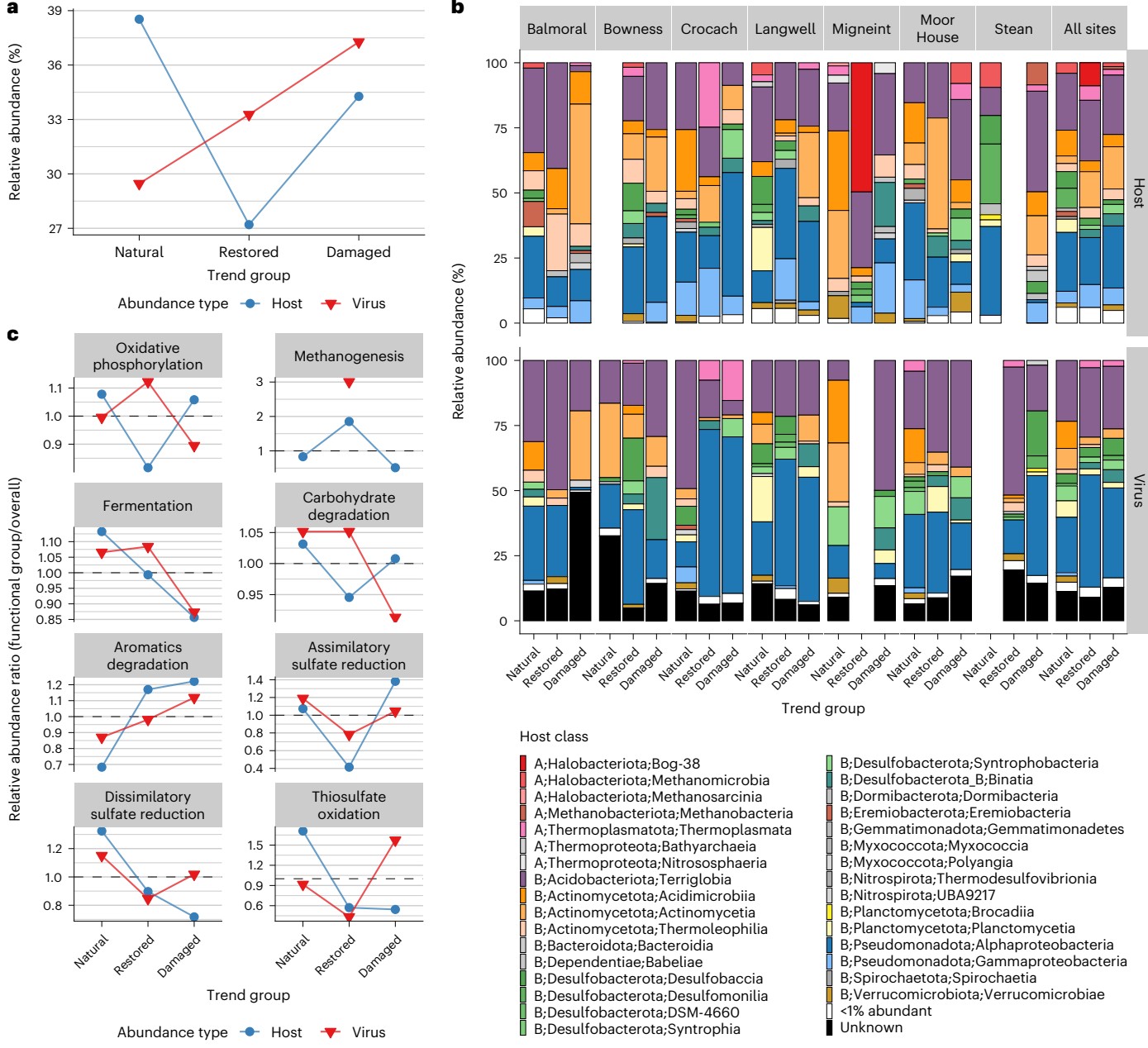

**Fig. 3 | Relative abundance of viruses and hosts across EHSs. a**, The relative abundance of virus (*n* = 1,448 genomes) and host (*n* = 411 genomes) genomes differentially abundant across EHS 'trend groups'. **b**, The relative abundance of host genomes (*n* = 411 host genomes, top row) and virus genomes (*n* = 1,351 virus genomes, bottom row), categorized by host class. Only differentially abundant host and viral genomes are included. Host classes are labelled as follows: Domain (A = Archaea, B = Bacteria); Phylum; Class. Individual classes that were <1% in relative abundance were grouped into the same '<1% abundant' category. **c**, Abundance ratios of differentially abundant hosts encoding eight key metabolic functions relevant to peat soils (*n* = 407 host genomes) and their predicted viruses (*n* = 550 virus genomes).

## Viral proteins are functionally distinct across EHSs

To assess viral functional differences across EHSs at the protein level, we clustered protein-coding viral genes from all sites and examined their distribution across EHSs (Fig. 4a). The three largest groups were protein clusters unique to individual EHS, suggesting that soils of each EHS harbour viruses encoding proteins with distinct functions. This degree of adaptation is notably greater than what we previously observed at the genome level (Fig. 2b). These patterns indicate that common pools of viral genomes exist across EHSs, but their functional potential is locally adapted to their specific environmental conditions. Despite these functional distinctions, the distribution of functional categories (based on PHROG[22]) across the intersections remains consistent

(Fig. 4a). Therefore, while viruses are specialized at the protein level, they perform similar high-level functions across all EHSs. Overall, our results demonstrate that viral protein functions are finely tuned to their environments, even when broader categories are conserved.

We also focused on viral AMGs[23] and their distribution across EHSs (Fig. 4b). AMGs are host-derived proteins with metabolic functions that provide viruses with evolutionary and fitness benefits. Similar to the all-protein results, the largest intersections correspond to KEGG[24] protein families unique to individual EHS, reinforcing the idea that the metabolic functions encoded by these viral genomes are distinct across different environmental conditions. Likewise, the distribution of high-level KEGG metabolism categories across the major intersections

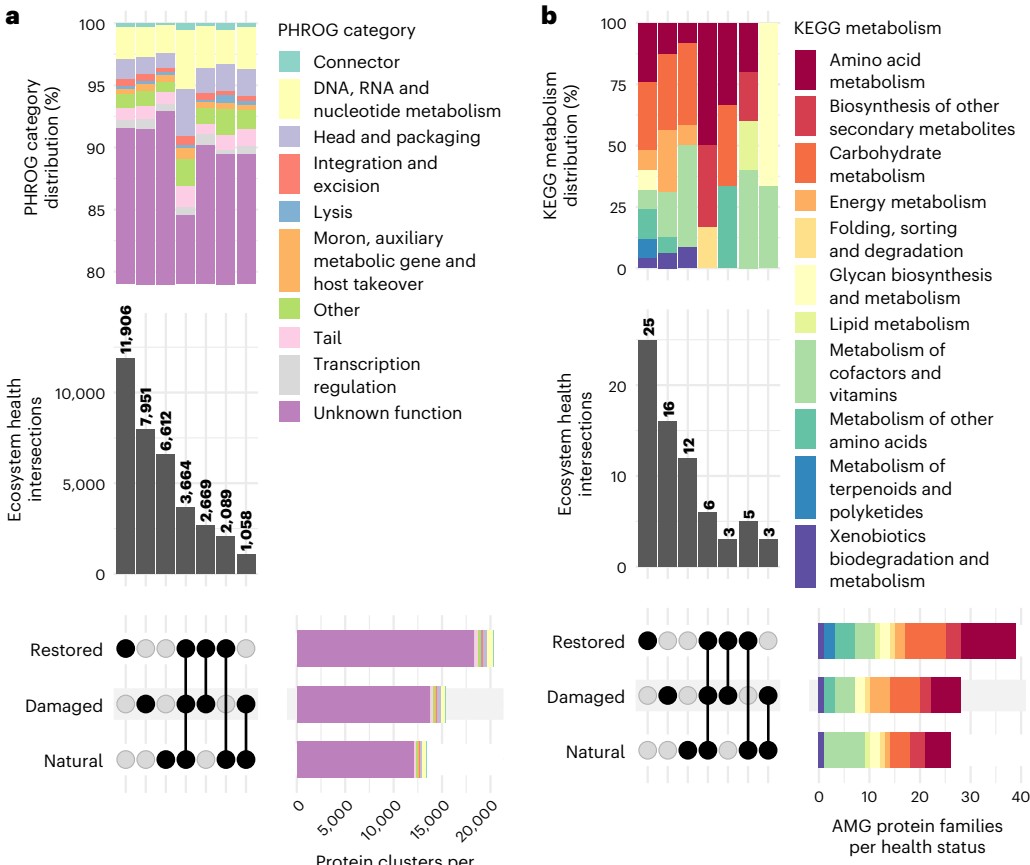

**Fig. 4 | Viral protein-coding genes and AMGs across EHSs. a**, UpSet plot showing amino acid identity-based clustering of all viral protein-coding genes ($n$ = 77,662 genes). Intersections represent protein clusters with viral proteins from multiple EHSs, while non-intersecting groups represent proteins unique to a single EHS. The distribution of PHROG[22] functional categories across these intersections is shown in the stacked bar plot at the top. **b**, UpSet plot of unique KEGG KOfams[24] ($n$ = 59 families) among viral AMGs ($n$ = 100 genes) across EHSs. Intersections indicate KOfams shared across different EHSs, while non-intersecting groups highlight KOfams unique to a single EHS. The stacked bar plot at the top illustrates the distribution of KEGG metabolism categories associated with these viral KOfams across the intersections.

remained largely similar, with categories such as 'Carbohydrate metabolism,' 'Metabolism of cofactors and vitamins' and 'Amino acid metabolism' being well represented. Yet, there was a small increase in the proportion of energy metabolism genes in the damaged-only samples compared with the natural-only and restored-only samples, with predicted functions involved in sulfur metabolism (K20034 3-(methylthio)propionyl-CoA ligase), methane metabolism (K16370 6-phosphofructokinase 2 and K15229 methylamine dehydrogenase heavy chain) and oxidative phosphorylation (K02107 V/A-type H[+]/Na[+]-transporting ATPase subunit G/H). This subtle shift may indicate functional adaptation, with viruses in damaged, oxygenated soils potentially playing a more active role in processes linked to electron transport in their hosts for their selfish benefit[25,26]. Altogether, viral proteins and AMGs are not distinct at high-level functions across EHSs, yet they are locally adapted to specific restoration contexts.

### Virus–host infection dynamics change with EHS

Viruses are dependent on their hosts to replicate, but their modes and rates of replication vary[27,28]. To this end, we next investigated virus–host infection dynamics using genome abundances of our bioinformatically predicted virus–host pairs. Linear regressions between total virus abundance and total host abundance across EHSs reveal complex interactions that vary by phylum (Fig. 5a). Notably, while the slopes of these regressions change within each phylum depending on EHS, all slopes are consistently less than 1. For example, in Acidobacteriota,

the slopes are 0.567 ($R^2$ = 0.84, BH-adjusted $P$ = 2.67 × 10^{-7}, $n$ = 18 soil samples) in natural soils, 0.812 ($R^2$ = 0.55, BH-adjusted $P$ = 3.81 × 10^{-4}, $n$ = 19 soil samples) in restored soils and 0.719 ($R^2$ = 0.84, BH-adjusted $P$ = 3.19 × 10^{-6}, $n$ = 15 soil samples) in damaged soils, indicating that host genomes are generally more abundant than their associated viral genomes across all EHSs at the phylum level. This pattern suggests chronic or non-lytic modes of infection at high host densities, known as 'piggyback-the-winner' dynamics[28,29], where viruses coexist with their hosts through non-lethal replication strategies, such as lysogeny, involving integration into the host genome. This pattern was also observed for viruses and hosts of other dominant phyla, but the strengths of these relationships were susceptible to changes in EHS (see Supplementary Results).

### Lineage-specific shifts in lysogeny and induction across EHSs

While 'piggyback-the-winner' dynamics prevailed in UK peatland soils, patterns of temperate (hereafter lysogenic) and actively replicating viral abundance across EHSs highlighted significant shifts in virus–host interactions. We identified 297 lysogenic viruses in total, 13% of all identified viruses, and analysed their abundances in each sample. Patterns of lysogenic virus abundance varied across sites, with no significant differences in their raw mean abundances when aggregating all sites (Extended Data Fig. 5). However, when normalizing lysogenic virus abundance by the total virus population in each sample (Fig. 5b), we found that the proportion of lysogenic viruses was significantly lower

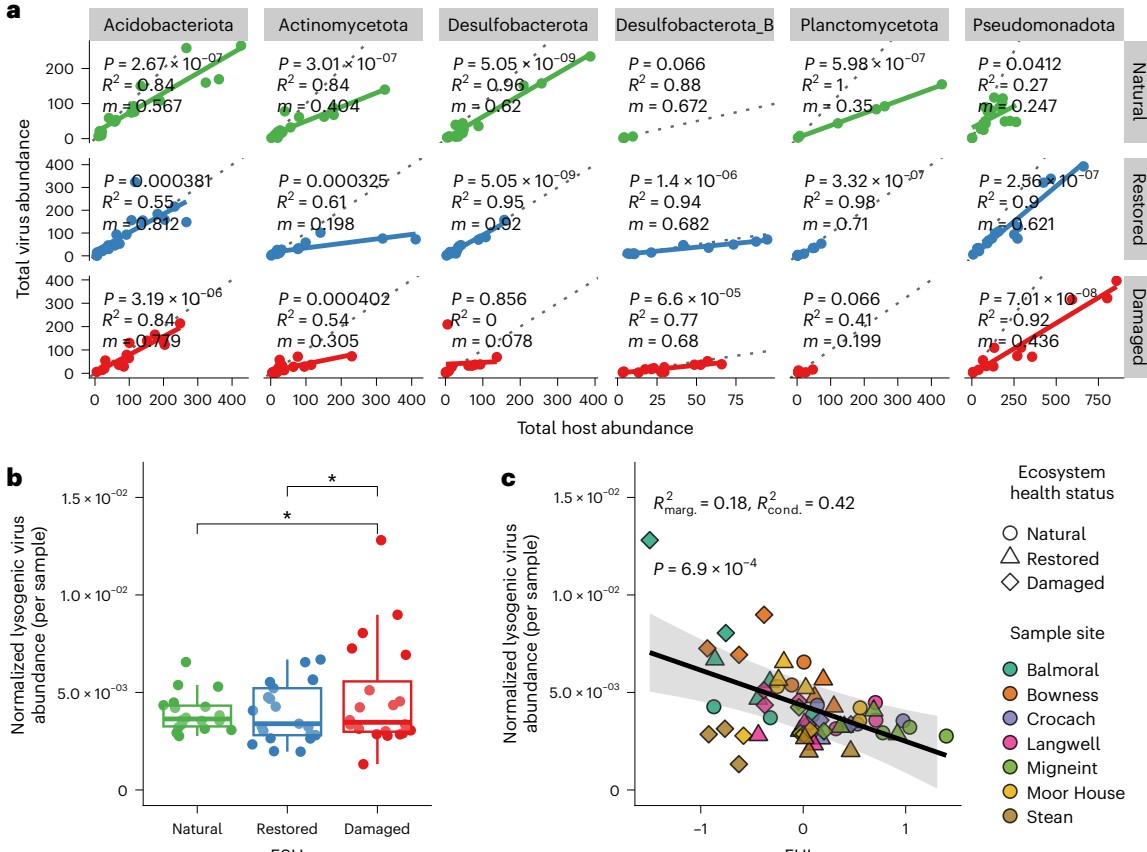

**Fig. 5 | Dynamics of virus–host interactions across EHSs. a**, Linear regressions of virus and host abundances by EHS. For clarity, only the six host phyla with the highest number of observations are shown. Linear model statistics (two-sided) for each phylum × EHS combination are provided, where $m$ is the slope of the best-fit line (shown as a solid line representing the fitted regression mean) and $P$ gives BH-adjusted $P$ values for the significance of each regression slope. Dotted lines represent a hypothetical slope of $m = 1$. **b**, Average trimmed mean genome coverage of lysogenic viruses per sample (normalized by the total trimmed mean coverage of all virus genomes; $n = 60$ soil samples) across all sites, grouped by EHS. Significant pairwise contrasts among EHSs are shown (estimated marginal means, two-sided, *$P \le 0.01$, BH-adjusted), determined from a linear mixed-effects model with sample site as a random intercept. Boxplots: centre line, median; box limits, upper and lower quartiles; whiskers, 1.5× interquartile range; points, individual data points. **c**, Linear mixed-effects model predicting normalized lysogenic virus abundance per sample in **b** ($n = 60$ soil samples) from EHI, with site as a random intercept. The black line shows the marginal fitted values (population-level mean predictions) from the linear mixed-effects model, and the shaded band represents the corresponding 95% confidence intervals. The marginal ($R^2_{marg.} = 0.18$) and conditional ($R^2_{cond.} = 0.42$) $R^2$ of the model fit are shown, and $P = 6.9 \times 10^{-4}$ (unadjusted) reflects the result of a Type II ANOVA assessing the significance of EHI as a fixed effect in the model.

in natural and restored soils compared with damaged soils (estimated marginal means, BH-adjusted $P = 0.0300$ and $P = 0.0398$, respectively). This suggests that lysogenic viruses contributed more substantially to the overall viral community in damaged soils. Furthermore, when modelling normalized lysogenic virus abundance as a function of the EHI while accounting for site-level variation (Fig. 5c), we observed a significant negative relationship (marginal $R^2 = 0.18$, conditional $R^2 = 0.42$, $\chi^2 = 11.52$, BH-adjusted $P = 6.9 \times 10^{-4}$, Type II ANOVA, $n = 60$). This indicates that the relative abundance of lysogenic viruses increases with peatland degradation. Together, these findings suggest an increase in the replication of lysogenic viruses as peatlands shift from natural to damaged states.

We aimed to identify actively replicating viruses in our samples by calculating virus-to-host abundance ratios (also known as virus:microbe ratio, or VMR) (Extended Data Fig. 6). We considered a virus to be 'active' if the virus:host ratio exceeded 10. Using this threshold, we identified 51 active viruses across 46 samples. This represented 10% of all viruses with host predictions and non-zero virus and host abundances. Of the 51 active viruses, 27 (53%) were also predicted to be lysogenic, accounting for 9.1% of all predicted lysogenic viruses. Thus, these active lysogenic viruses probably underwent recent induction at

the time of soil sampling. Among them, 26% were active in natural soils, 41% in restored soils and 67% in damaged soils (13 lysogenic viruses were active in more than one sample, explaining why the total exceeds 100%). We also found that EHS had a significant effect on virus:host ratios, but the effects varied by the host family (see Supplementary Results). In summary, these results support our observation that both overall viral genome abundance and the proportion of lysogenic virus genomes increase in damaged soils, with a subset of these viruses probably undergoing greater induction and replication compared with those in natural and restored peatlands.

## Discussion

Peatlands are the world's largest terrestrial carbon stores[1–4] but are increasingly threatened by habitat destruction, shifting from being carbon sinks to becoming carbon sources[5–9]. Since carbon cycling in peatlands is primarily driven by soil microorganisms[3,13], understanding how environmental damage and restoration affect soil microbiomes is crucial for managing peatlands and mitigating their carbon emissions. Here we show that restoration of peatland ecosystem health (1) significantly shaped viral community composition, (2) enriched viruses infecting specific microbial lineages and functional groups, (3) selected

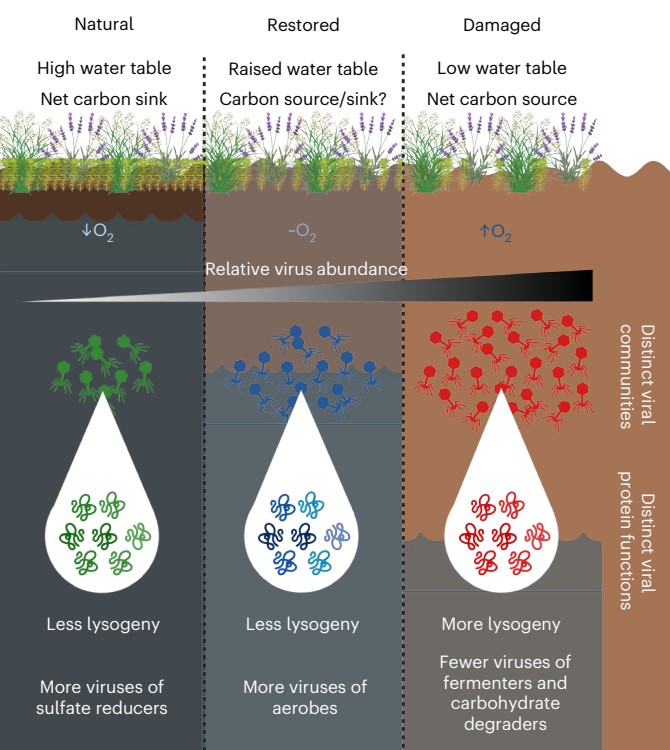

**Fig. 6 | Summary of dynamic viral communities across an ecosystem health gradient in peatland topsoils.** Along a gradient from natural peatlands (high water table, anoxic, net carbon sink) to restored and damaged sites (low water table, more oxic conditions, net carbon source), viral relative abundance, infection strategy (lysogeny) and host associations shift. These changes lead to distinct viral communities and protein functions, and are met with enrichment of viruses of sulfate reducers in natural soils, aerobic hosts in restored soils and depletion of viruses that infect fermenters and carbohydrate degraders in damaged soils. Illustration created with BioRender.com.

for distinct viral protein functions and (4) altered virus–host population dynamics, advancing our understanding of how environmental change impacts soil viruses and their roles in global carbon cycling (Fig. 6).

We found that viral abundance and composition often diverge from those of their microbial hosts across EHSs, rather than mirroring host populations. This decoupling, particularly notable in carbon- and sulfur-cycling hosts, suggests that viral responses are influenced by factors beyond host availability, potentially by environmental stressors such as nutrient shifts[30] or soil chemistry changes[15], or by host physiological responses affecting susceptibility to infection[30]. In parallel, viral proteins showed local adaptation to ecosystem health, with distinct metabolic functions detected in damaged soils, including AMGs involved in methane metabolism, oxidative phosphorylation and sulfur metabolism. However, our metagenomic approach captures only potential functions, and functional assays or transcriptomics are needed to clarify the impact of these viral adaptations on ecosystem recovery.

Restoration also shifted viral replication strategies, with an increase in lysogeny and increased activity among a subset of viruses in damaged soils. This aligns with 'piggyback-the-winner' dynamics[28,29], and is in line with microbial studies showing that damaged peatlands have higher microbial growth rates and population sizes[18], and that environmental changes can induce switches in virus lifestyle[20,31–33]. These results highlight the sensitivity of soil viral communities to environmental disturbances, and suggest that shifts in viral replication strategies could serve as indicators of host population densities and EHS in peatlands undergoing restoration. Microbial metagenomic

approaches are often biased towards viruses in an intracellular state[34,35], but despite this limitation, current bioinformatic tools can recover most environmental viruses from metagenomes[36,37] and increasingly offer reliable host range predictions[38].

As peatlands are restored to mitigate climate change, understanding virus–host interactions is essential not only for predicting microbial responses, but also for identifying how viral regulation of host populations and metabolism may influence the recovery of carbon storage and other ecosystem functions. Our findings suggest that viruses do not simply track host populations but actively respond to environmental conditions associated with degradation and restoration. Such responses may influence microbial turnover rates, metabolic activity and biogeochemical cycling as they have in other soil ecosystems[39–41], all of which are critical for peatland functioning as net carbon sinks. Therefore, viruses probably play an underappreciated role in shaping restoration trajectories. Future studies should verify these findings with functional and experimental approaches, particularly focusing on viral influences over key microbial functional groups. Integrating viral community dynamics into restoration monitoring will strengthen our ability to assess and enhance peatland ecosystem recovery.

## Methods

### Soil sampling and metagenome sequencing

Soil samples were collected between May and October 2021 from seven upland peatland sites across Britain covering a gradient of climatic conditions[18]. At each site, we sampled three areas with different ecosystem EHSs: a near-natural reference (natural), damaged by drainage or erosion (damaged) and restored by rewetting through drain blocking (restored). Using a standard soil corer or a Russian peat corer, three replicates were sampled per EHS, with the replicates being locally adjacent to minimize the impact of underlying geology and climatic conditions. Each replicate was sampled 5 m apart across a 10-m transect after removing surface vegetation. Most sites underwent restoration within the past 10 years. The exact duration since drainage is unknown, but it probably spanned several decades. Samples from damaged and restored areas were taken 2 m away from drainage features. Samples from each EHS were taken from areas with similar peatland lawns. Overlaying peatland land cover data were obtained from ArcGIS Hub at hub.arcgis.com/datasets/Defra::peaty-soils-location-england (England), hub.arcgis.com/datasets/theriverstrust::unified-peat-map-for-wales (Wales) and hub.arcgis.com/datasets/snh::carbon-and-peatland-2016-map (Scotland). See Supplementary Table 1 for sample metadata and locations.

Total community DNA was extracted from 0.25-g aliquots of homogenized soil collected from the upper 10-cm layer of samples, using the DNeasy PowerSoil Pro kit (Qiagen) following manufacturer instructions. DNA concentration and integrity were evaluated using Nanodrop spectrophotometry and Qubit fluorometric assays. Library preparation was performed with the NEBNext Ultra II FS DNA kit following manufacturer guidelines. Sequencing was conducted on an Illumina NovaSeq platform at the NERC Environmental Omics Facility[18].

### Analysis of soil environmental parameters

Oxygen concentrations were measured at depths of 0–5 cm and 5–10 cm using a fibre optic oxygen sensor (OXROB10, PyroScience); we then averaged the two measurements for each sample and used the mean in all subsequent analyses. Moisture content was determined gravimetrically and expressed as a percentage of the total mass. pH and soil conductivity were measured from a slurry prepared by mixing 5 g of peat with 25 ml of deionized water. Soil samples were dried, ball milled and subsampled, with 10–12 mg weighed into tin capsules for total carbon and total nitrogen measurements, obtained using an NA 2500 Series elemental analyser (CE Instruments). All environmental measurements were obtained from the same sample aliquots and consolidated into a single dataset keyed by sample identifier and annotated

with sampling site and EHS (Supplementary Table 1). For subsequent analyses, soil environmental parameters were either globally scaled (centred and scaled across all sites) or site-specifically scaled (centred and scaled within each site), depending on whether we aimed to emphasize across-site variability or within-site variability.

Because some site × EHS combinations had missing values for at least one environmental parameter, we used a mixed-effects modelling approach to impute these data before PCAs. Specifically, for each variable with missing values, we fitted a linear mixed-effects model with EHS as a fixed effect and site as a random intercept, and used the resulting model to predict missing observations. This allowed us to retain samples that would otherwise be omitted while preserving site- and EHS-specific trends. Oxygen concentration (10/66 samples), pH (6/66) and conductivity (6/66) required imputation; the globally scaled dataset was used when imputing for the overall PCA, and the site-specific dataset was used for each site-specific PCA. No other parameters required imputation. PCAs were then conducted on the imputed datasets using the 'prcomp' function in R. We performed one PCA including all samples (globally scaled) and separate site-specific PCAs (site-specifically scaled).

### Sequencing read quality control and metagenome co-assembly

Metagenome sequence reads underwent quality control, filtering, assembly and formatting using the Anvi'o v.8 metagenomics workflow[42]. Raw sequence reads were quality filtered with Illumina-utils (v.2.13)[43]. Filtered read libraries, generated in triplicate from the same sample site × EHS combinations (Supplementary Table 1), were co-assembled into metagenomes using MEGAHIT (v.1.2.9)[44], utilizing the 'meta-large' preset to optimize $k$-mer selection for large complex metagenomes such as those found in soil. A minimum contig length of 1 kb was enforced. Metagenome assembly statistics were evaluated with metaQUAST (v.5.2.0)[45], and filtered reads were mapped back to their respective metagenomes using Bowtie2 (v.2.5.1)[46] to assess read recruitment (Supplementary Table 2).

### Host genome binning, quality control and taxonomic assignment

For each metagenome co-assembly, contigs were binned into MAGs using MetaBAT 2 (v.2.15)[47], utilizing the metagenome read-mapping files described above to aid in binning. A minimum percent identity of 97% and a contig length of at least 1 kb for mapped reads were required. Binning was performed with a minimum contig size of 2.5 kb, and default MetaBAT 2 parameters were applied. The completeness and contamination of the bins were assessed with CheckM (v.1.2.2)[48], using the lineage workflow. On the basis of CheckM results, bins were categorized into high-quality (≥90% completeness and ≤10% contamination), medium-quality (≥50% completeness and ≤10% contamination) and low-quality (<50% completeness or >10% contamination) MAGs. Taxonomic assignments for medium- and high-quality MAGs were determined using the GTDB-tk v.2.3.2 de novo workflow[49]. Patescibacteria and Altiarchaeota were selected as the bacterial and archaeal outgroups for phylogenetic tree inference, as the CheckM marker gene lineage results indicated that these phyla were underrepresented among the medium- and high-quality MAGs. Only high- and medium-quality MAGs were included in subsequent analyses involving host genomes.

### Viral sequence identification, binning, host prediction and species cluster formation

ViWrap (v.1.3.0)[50] was used to process each metagenome co-assembly, running on the metagenome contigs along with their associated triplicate filtered read pairs. The parameters '–identify_method genomad' and '–input_length_limit 2000' were specified to identify viral contigs using GeNomad (v.1.7.4)[37] and to enforce an initial minimum viral contig length of 2 kb. ViWrap utilized Bowtie2 v.2.4.5 in 'end-to-end' mode

to map each filtered read pair to the viral contigs identified by GeNomad, generating the necessary coverage files for binning. ViWrap then binned viral contigs into vMAGs using vRhyme (v.1.1.0)[51] with multisample read coverage statistics. Both binned viral contigs and unbinned single-contig viral genomes are hereafter referred to as vMAGs.

Upon completion of ViWrap for each co-assembly, vMAGs and their summary information were extracted and renamed using custom Python scripts. Host genomes and taxonomy for all generated vMAGs were predicted with iPHoP (v.1.3.3)[38], using a custom host genome database that included both the default 'iPHoP_db_Aug23_rw' genomes and the high- and medium-quality host MAGs described earlier. To ensure that iPHoP did not treat individual contigs in multicontig vMAGs as separate genomes, these contigs were linked by sequences of 1,500 Ns using the vRhyme auxiliary script 'link_bin_sequences.py', and the iPHoP parameter '–no_qc' was used to prevent N-linked vMAGs from being discarded. Host predictions with a minimum confidence score of 90% were retained using the default iPHoP parameter '–min_score 90'.

vMAGs from all co-assemblies were dereplicated into viral 'species'-level clusters using dRep (v.3.5.0)[52]. A minimum representative genome size of 5 kb was enforced with the parameter '-l 5000'. The parameters '–ignore_genome_quality -pa 0.8 -comW 0 -conW 0 -strW 0 -N50W 0 -sizeW 1 -centW 0' were applied as recommended by the dRep documentation for non-bacterial/archaeal genomes. In addition, the parameters '-sa 0.95' and '-nc 0.85' were used to form species clusters at 95% average nucleotide identity (ANI) with a minimum aligned coverage of 85%, employing skani (v.0.2.1)[53] for genome comparisons.

### vMAG genome clustering with soil viral genome databases

To assess how well the vMAGs generated in this study are represented among other described soil viral genomes, we obtained viral genomes from publicly available soil virus databases. To ensure a comprehensive collection, we selected three databases: PIGEON (v.1)[17] (filtered to include only viral genomes assembled from soil samples), the Global Soil Virome[20] and the Global Soil Virus Atlas[19]. We clustered the vMAGs with viral genomes from these databases on the basis of amino-acid identity (AAI). Protein-coding genes in all viral genomes were predicted and translated using pyrodigal-gv (v.0.3.1)[37,54] (github.com/althonos/pyrodigal-gv). Pairwise AAI measurements were obtained by first creating a protein sequence database with MMseqs2 (v.15.6f452)[55], followed by running mmseqs search with the protein database against itself. This was done with a minimum amino-acid sequence identity of 0% to retain all possible pairwise comparisons (mmseqs parameter –min-seq-id 0.0) and a minimum alignment coverage of 30% (mmseqs parameter -c 0.3). The resulting pairwise AAI measurements were computed and used to form approximate genus-level genome clusters using custom Python scripts. See Supplementary Methods for details.

### vMAG and host MAG abundance, coverage estimation and presence/absence

To perform differential abundance and beta-diversity analyses of microbial communities from metagenomes, it is essential to use 'species' counts from a non-redundant set of taxa[21]. Filtered metagenome reads were mapped to the dereplicated, species-representative vMAGs using Bowtie2 v.2.4.5 with the '–sensitive' parameter in 'end-to-end' mode. Read-mapping files were then sorted and indexed using SAMtools (v.1.17)[56]. Following community-established standards[57], read-mapping files were filtered to remove reads with <90% identity using CoverM (v.0.6.1)[58]. CoverM was also used to generate three tables for the analysis of species-representative vMAGs abundance, for each metagenome read sample, as is common in other viral community studies[17,35,41]: (1) absolute mapped read counts, (2) trimmed mean genome coverages (with the top and bottom 5% of covered bases removed) and (3) genome coverage fraction (also known as 'breadth'). A minimum coverage fraction of 0 was used in generating each table.

Host MAGs were dereplicated using dRep, with the only changes to default parameters being the use of '–ignoreGenomeQuality' (since quality had already been assessed) and '–S_algorithm skani' to use skani for genome comparisons. Filtered metagenome reads were mapped to the species-representative host MAGs, and abundance and coverage statistics were generated using the same tools and parameters applied to the vMAGs.

To assess the distribution of species-representative viral genomes across sample sites and EHSs, we used a minimum genome breadth of 50% to consider a viral genome as present in a given sample.

### Statistical analyses of viral and host community composition

We assessed viral and host community composition across EHSs by calculating Bray–Curtis dissimilarities from normalized, species-representative genome coverage data, followed by PCoA. We restricted viral community analyses at the Langwell site to replicates with the longest post-restoration duration. A minimum genome breadth of 0.50 was used to filter genome abundances before analysis to avoid false positives. We tested for separation of samples by sampling site and by EHS using ANOSIM. To identify ecological drivers of viral community structure, we performed permutational multivariate analysis of variance (PerMANOVA) with site as a blocking factor, testing contributions from host community composition (via host PCoA axes), EHS and a continuous EHI. We confirmed homogeneity of dispersion before all PerMANOVA analyses. Variance partitioning and distance-based redundancy analyses (dbRDA) were used to quantify the relative contributions of host composition, site and EHS to viral community structure. See Supplementary Methods for more detail on statistical analyses of viral and host community composition.

### Viral and host genome differential abundance and EHS group assignment

The table of absolute mapped read counts for species-representative viral genomes, generated as described above, was used as the input for differential abundance analysis. Normalization was not performed, following software recommendations[21]. The genome count table was then split by sample site and used for differential abundance analysis with the R package DESeq2 (v.1.44.0)[21], performed separately for each sample site. Sample was included as a factor in the negative binomial generalized linear models fitted with DESeq2, using a likelihood ratio test to compare the full model (including EHS) to a reduced model (intercept only). $P$ values from these tests were adjusted using the false-discovery rate (FDR) method, with a maximum FDR-adjusted $P$ value of 0.05 to infer viral genomes that were differentially abundant across EHSs at each site. This workflow was also applied to host genome counts to identify differentially abundant host genomes.

To determine which of the differentially abundant viral and host genomes were enriched in soils corresponding to each EHS, we performed hierarchical clustering of their trimmed mean genome coverages in R, following similar approaches used in past viral community ecology studies that analyse abundance patterns across groups[17,35]. For each sample site, the same normalized trimmed mean genome coverages used in the community composition analyses were filtered to include only the differentially abundant viral/host genomes. These filtered trimmed mean coverages were converted into relative abundances (relative to the total abundance within each sample). $Z$-scores were calculated for the relative abundances of each viral/host genome in each sample, and the mean $z$-score was calculated for each genome across the different EHSs. Euclidean distances were calculated from the resulting mean $z$-scores, with 'NA' or missing values set to zero to maintain compatibility for clustering. The resulting viral and host distance matrices were hierarchically clustered using the R function 'hclust' with the 'ward.D' method. The cluster trees were cut into three groups, as there were three EHSs. The previously calculated mean $z$-scores for each viral/host genome in each EHS were plotted for the

three groups (Extended Data Figs. 3 and 4). These plots for each sample site were then inspected to assign the viral/host genomes in each group to one of three ecosystem health trend groups (EHS groups): 'Natural-enriched', 'Restored-enriched' or 'Damaged-enriched', on the basis of their abundance patterns.

### Host MAG metabolic function predictions

Putative metabolic functions encoded by host MAGs were predicted using METABOLIC (v.4.0)[59]. We focused on eight functions relevant to peatland soil ecosystems, including oxidative phosphorylation, methanogenesis, fermentation, carbohydrate degradation, aromatics degradation, assimilatory and dissimilatory sulfate reduction and thiosulfate oxidation. Function presence was inferred from KEGG module and functional annotations reported by METABOLIC-C. For full pathway definitions, the justification for their inclusion and KEGG module-level criteria, see Supplementary Methods.

### Viral and host relative abundance across EHS groups

We quantified virus and host genome relative abundances across EHSs using normalized genome coverage data, filtered to retain differentially abundant genomes assigned to one of the three EHS groups. Relative abundances were compared across host class and groups of host genomes encoding the metabolic functions identified above, with enrichment of a metabolic function assessed by calculating relative abundance ratios of genomes encoding the function normalized to overall viral and host abundance in each EHS group. For full details on relative abundance calculations across EHS groups, see Supplementary Methods.

### Total virus over total host abundance regressions

To analyse the relationship between the abundance of viruses and their hosts across EHSs, normalized trimmed mean genome coverages of vMAGs and host MAGs from all sample sites were filtered to include only predicted virus–host pairs and only values >0. In addition, we required that both viruses and hosts were assigned to the same EHS group within a given sample to ensure that abundance comparisons accurately reflected shared ecological contexts and trends, preventing potential bias from mismatched EHS dynamics. The remaining data were then summarized at the host phylum level to provide a broad overview of virus–host relationships for specific host lineages. Specifically, the total abundance of hosts within each phylum and their associated predicted viruses were calculated for each sample, site and EHS combination. Linear models of virus-to-host abundance relationships were fitted for each host phylum–EHS combination using the R function 'lm', adjusting the resulting $P$ values using the Benjamini–Hochberg (BH) method.

### Lysogenic virus abundance and statistical analysis of active viruses

To assess temperate (hereafter lysogenic) virus population dynamics across EHSs, we identified temperate phages using the classifications provided by ViWrap (see github.com/AnantharamanLab/ViWrap#notes and github.com/AnantharamanLab/vRhyme#interpreting-vrhyme-binsvmags-) and normalized their abundances by the total virus abundance per sample. Differences in normalized lysogenic virus abundance across EHSs and its relationship with a continuous EHI were evaluated using linear mixed-effects models. Viral replication activity was estimated by calculating virus-to-host abundance ratios, with active viruses defined as those exceeding a 10:1 ratio. See Supplementary Methods for more detail on our statistical analyses of lysogenic virus abundance.

### vMAG protein clustering

To assess the distribution of homologous proteins across EHSs, we clustered the translated amino acid sequences of all protein-coding viral

genes, obtained as described previously (see 'vMAG genome clustering with soil viral genome databases'). Protein clustering was performed with MMseqs2 v.15.6f452 using the 'mmseqs cluster' command and the parameters '–cluster mode 0 –cov-mode 0 -s 7.5' as well as '–min-seq-id 0.25 -c 0.5' to enforce a minimum sequence identify of 25% and alignment coverage of 50% to ensure that alignments were representative of whole proteins rather than individual domains. The resulting protein cluster table was used to analyse the intersections of EHS membership for proteins within protein clusters as described below.

## vMAG protein functional annotations, AMG prediction and curation

An HMMsearch[60] was performed on all vMAG-encoded amino acid sequences using profile HMMs from multiple databases, including PHROGs (release 2022-01-17x)[22] and KEGG KOfam (March 2019 release)[24]. See Supplementary Methods for a full list of databases used, their versions and details on HMM searches. To identify putative AMGs encoded by vMAGs, we employed a conservative approach that utilized functional annotations and genomic context statistics to avoid false-positive predictions, following community standards[61]. Briefly, we searched protein functional annotations for metabolic functions, removed proteins with functions that are commonly misannotated as AMGs[17,61], and removed likely non-viral protein contamination. For more detail, see Supplementary Methods. Although AMG functional assignments were available from multiple databases, only the remaining filtered and curated AMGs with KEGG KOfam annotations were retained for analysis of intersections across EHSs. This decision was made to simplify visualization and because KEGG KOfams encompass broader functional categories.

## Reporting summary

Further information on research design is available in the Nature Portfolio Reporting Summary linked to this article.

## Data availability

All raw sequencing data are publicly available at the NCBI Short Read Archive under BioProject accession PRJNA1203648. Whole assembled metagenomic contigs as well as high-quality prokaryotic metagenome-assembled genomes are available at the NCBI WGS using the same BioProject accession. All viral metagenome-assembled genomes and prokaryotic metagenome-assembled genomes (medium and high quality) are publicly available on figshare at https://doi.org/10.6084/m9.figshare.28143446 (ref. 62). Source data are provided with this paper.

## Code availability

All scripts for data processing and visualization are available on GitHub at https://github.com/AnantharamanLab/UKPeatlandViruses (ref. 63).

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

## Acknowledgements

This research was supported by the National Science Foundation under grant no. DBI2047598 (to K.A.). A.A.M. and K.A. were supported by Human Frontier Science Program research grant RGP018/2024 and by the University of Aberdeen International Partnership funding within UKRI Biotechnology and Biological Sciences Research Council (BBSRC) remit. J.C.K. was supported by the National Science Foundation Graduate Research Fellowship Program under grant no. 2137424. W.P. was funded by UKRI Natural Environment Research Council (NERC) Scottish Universities Partnership for Environmental Research (SUPER) Doctoral Training Partnership (DTP). A.A.M. and W.P. received funding for sequencing from NERC Environmental Omics Facility (NEOF). We thank the landowners, site managers and researchers who enabled access to the field sites used in this study, as well as the insights and support provided by all current and past members of the Anantharaman and Malik labs. In particular, we thank K. Ball for assistance with statistical analyses and M. Langwig for data analysis recommendations.

## Author contributions

A.A.M., K.A. and J.C.K. conceived of and designed the overall study. W.P. and A.A.M. designed the sampling strategy, collected soil samples and recorded associated metadata. W.P. processed the soil samples for sequencing and chemical analyses and subsequently collected the resulting data. J.C.K. performed quality control, assembly and analysis of metagenomic data, conducted statistical analyses and deposited the metagenomic and environmental datasets publicly. A.A.M. developed and calculated the ecosystem health index. J.C.K. prepared the figures and tables and wrote the initial draft of the paper. J.C.K., A.A.M. and K.A. interpreted the results and revised the paper. A.A.M. and K.A. secured funding and provided resources for the project.

## Competing interests

The authors declare no competing interests.

## Additional information

**Extended data** is available for this paper at https://doi.org/10.1038/s41564-025-02199-x.

**Correspondence and requests for materials** should be addressed to Ashish A. Malik or Karthik Anantharaman.

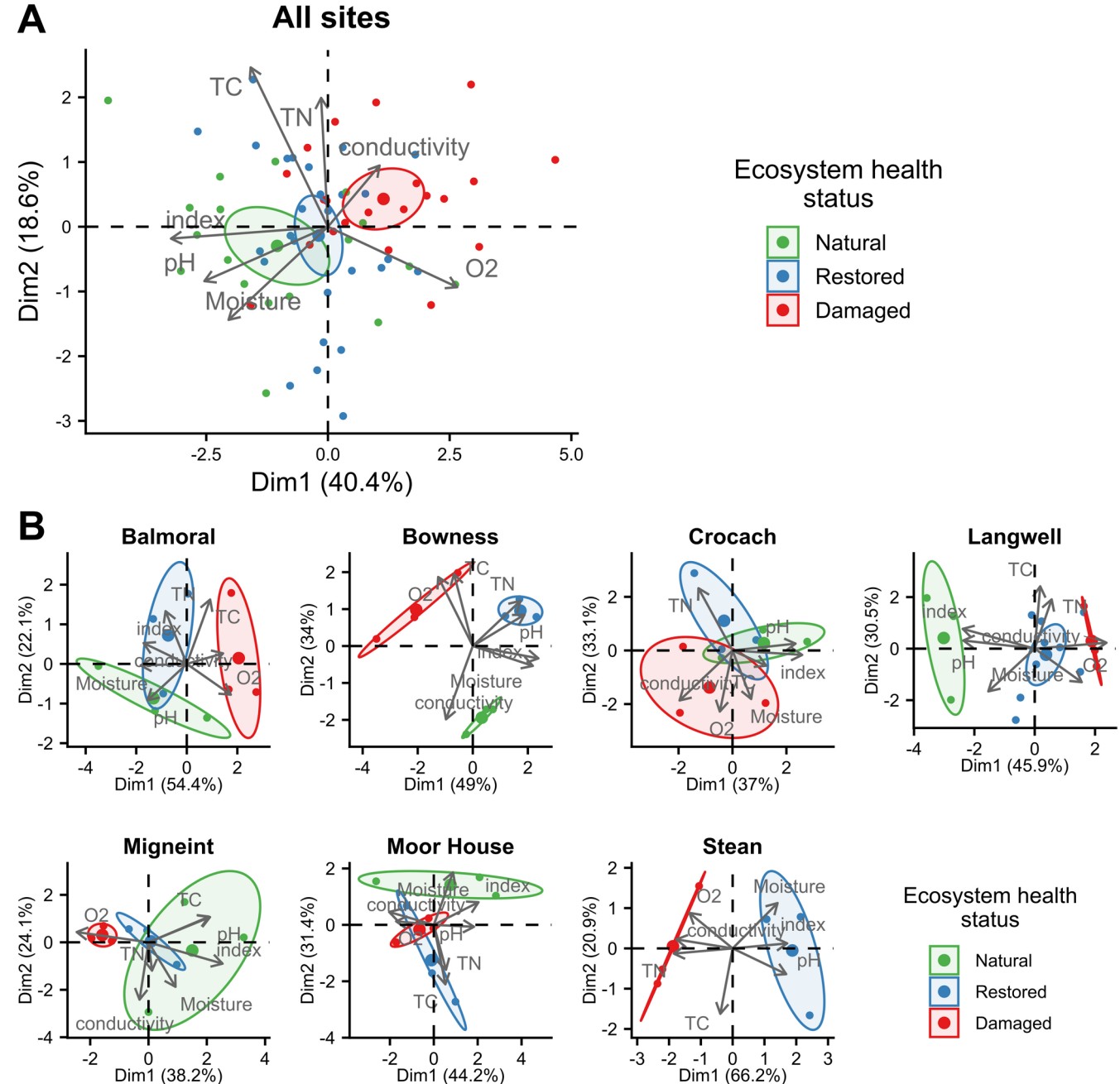

**Extended Data Fig. 1 | Environmental Composition of the Seven Peatland Sample Sites. (A)** Principal components analysis (PCA) of globally scaled environmental parameters for all samples, sites, and ecosystem health statuses (EHS; $n = 66$, missing measurements imputed [see Methods]). The percentages of variance captured by the first two PCA axes are provided. Each point represents an individual soil sample positioned according to its environmental parameter values, as summarized by the first two principal components. Ellipses, colored by EHS, represent 95% confidence intervals around the centroid of each group. Arrows indicate the direction and strength of each environmental parameter's contribution to the ordination, with longer arrows signifying stronger influence. **(B)** PCA for each sampling site, using data scaled within each site to emphasize local variation while minimizing broader site-level differences.

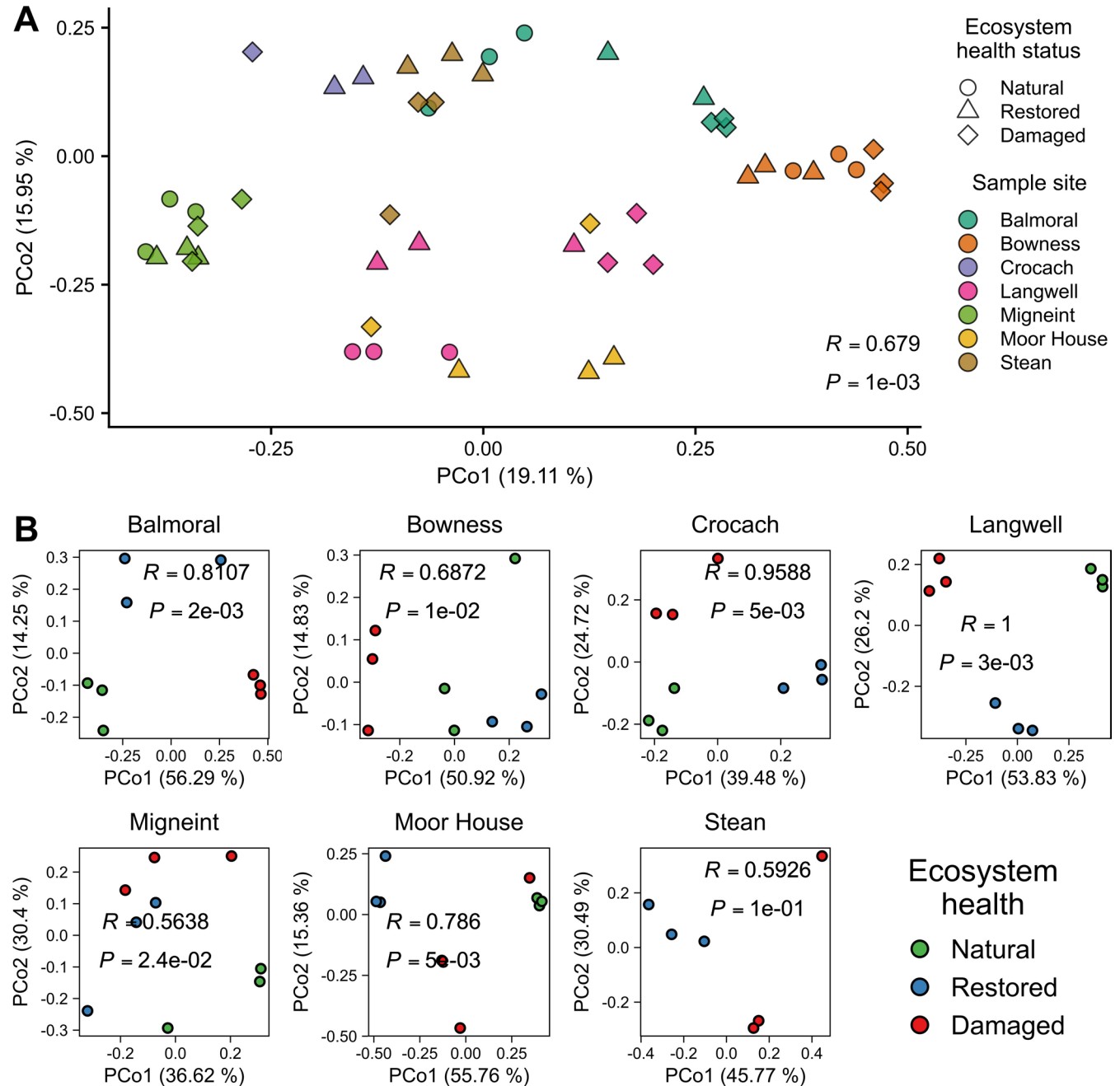

**Extended Data Fig. 2 | Bacterial and Archaeal Community Composition.**
(**A**) Principal coordinates analysis (PCoA) of Bray-Curtis dissimilarities of host metagenome-assembled genome (MAG) community composition across all sample sites. The percentages of variance captured by each PCoA axis are provided. Each point represents a host community of an individual sample.

Analysis of similarity (ANOSIM; 999 permutations) shows significant separation by site ($R = 0.679$, $P = 1.0e$-3, unadjusted). (**B**) PCoA of each sample site with ANOSIM statistics of ecosystem health status (999 permutations; exact $R$ and $P$ values shown on plots).

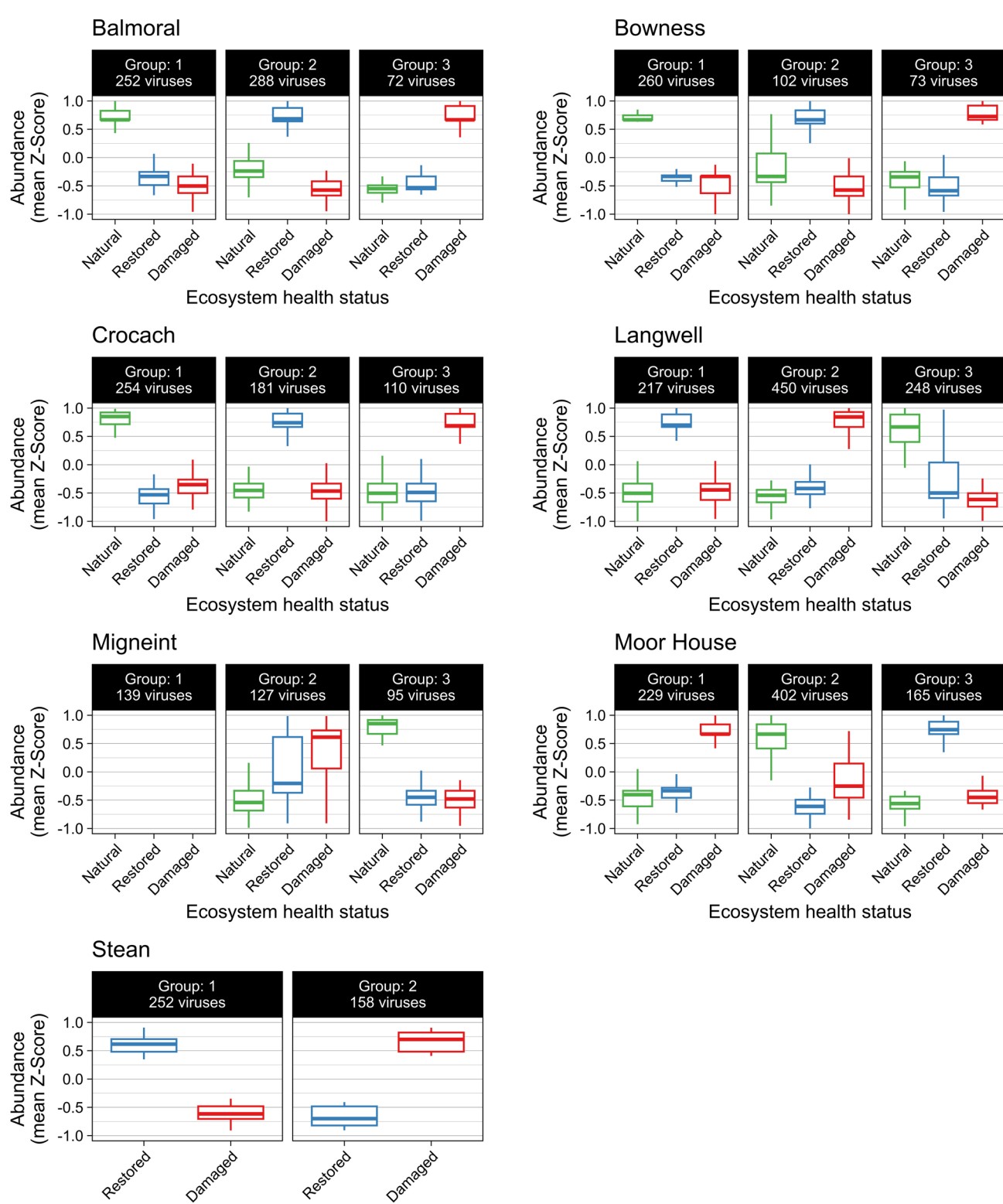

**Extended Data Fig. 3 | Average Z-scores of DESeq2-Significant Viral Genomes Across EHS.** Individual Z-scores for each virus genome were averaged in each ecosystem health status (EHS), separately, for each sample site. These trends were used to assign viruses to a particular EHS group. Groups without boxplots indicate groups with mean z-scores that are all invalid due to all group counts being 0. Viruses in these groups were assigned to the remaining EHS that was not assigned to the other groups in the same site. Boxplots: center line, median; box limits, upper and lower quartiles; whiskers, 1.5× interquartile range.

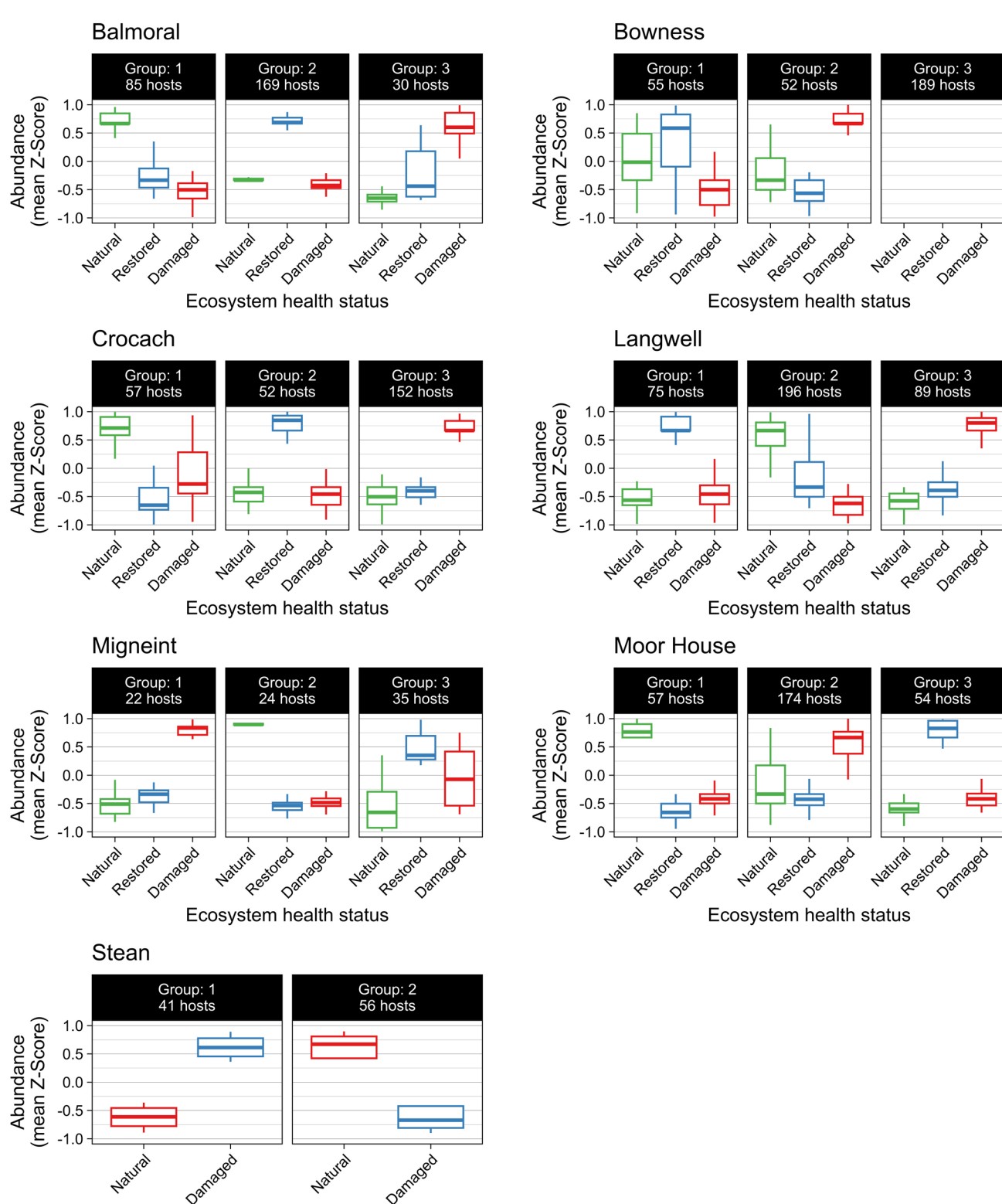

**Extended Data Fig. 4 | Average Z-Scores of DESeq2-Significant Host Genomes Across EHS.** Individual Z-scores for each host genome were averaged in each ecosystem health status (EHS), separately, for each sample site. These trends were used to assign hosts to a particular EHS group. Groups without boxplots indicate groups with mean z-scores that are all invalid due to all group counts being 0. Hosts in these groups were assigned to the remaining EHS that was not assigned to the other groups in the same site. Boxplots: center line, median; box limits, upper and lower quartiles; whiskers, 1.5× interquartile range.

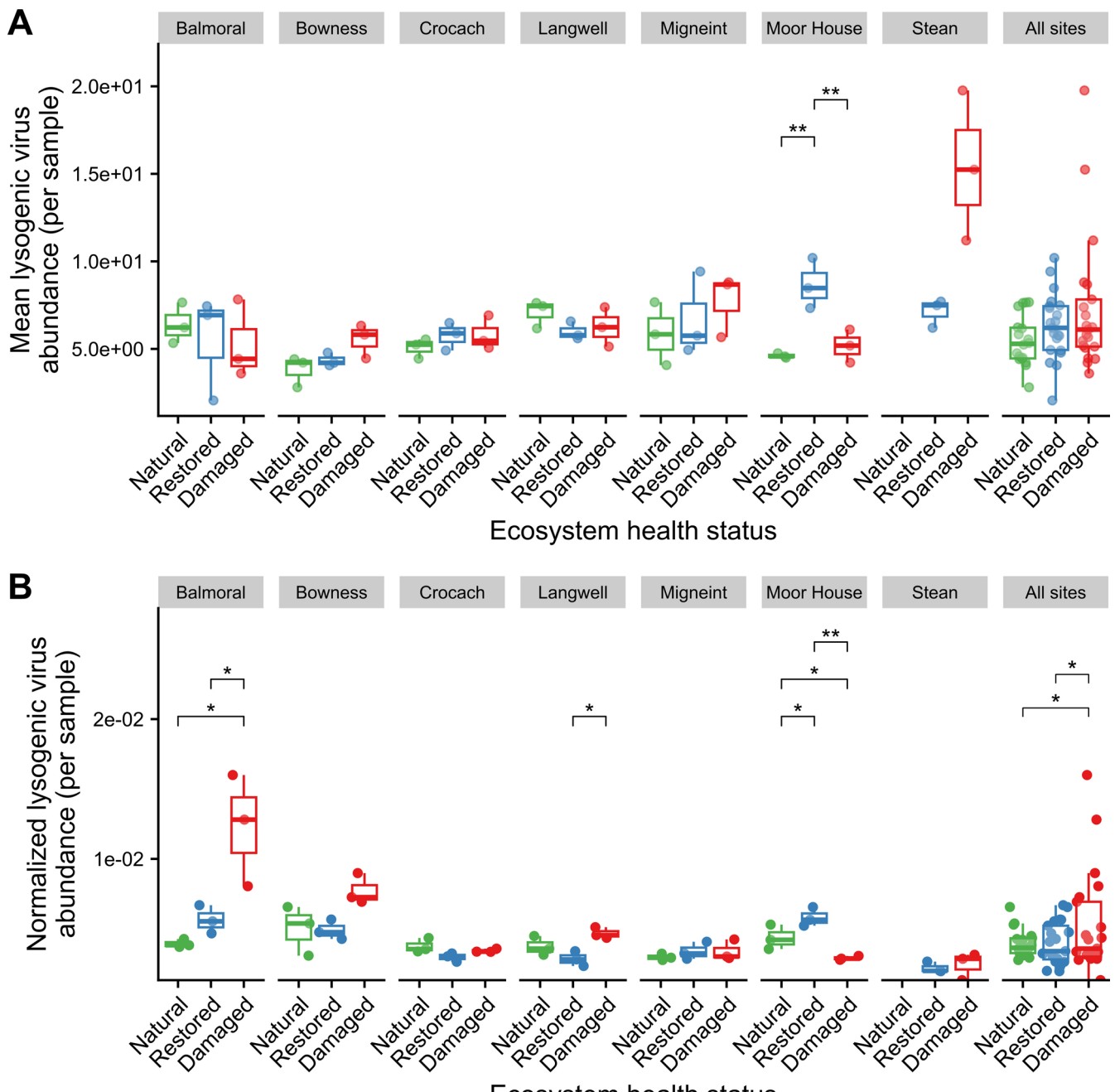

**Extended Data Fig. 5 | Abundance of Lysogenic Virus Genomes. (A)** Mean abundance of all lysogenic viruses (integrated prophage or contains an integrase/excisionase functional annotation, $n = 297$ genomes) per sample in each ecosystem health status (EHS) and sample site ($n = 60$ samples). Results for estimated marginal means (emmans, two-sided) pairwise contrasts among EHS in each site are provided (*$P \leq 0.05$; **$P \leq 0.01$, Benjamini-Hochberg adjusted, results with $P > 0.05$ are omitted). Boxplots: center line, median; box limits, upper and lower quartiles; whiskers, 1.5x interquartile range; points, individual data points. **(B)** The mean abundance of lysogenic viruses in **(A)**, normalized by the total virus abundance (lysogenic and non-lysogenic and unknown) in each sample. Results for estimated marginal means (emmans, two-sided) pairwise contrasts among EHS in each site are provided (*$P \leq 0.05$; **$P \leq 0.01$, Benjamini-Hochberg adjusted, results with $P > 0.05$ are omitted). Boxplots: center line, median; box limits, upper and lower quartiles; whiskers, 1.5x interquartile range; points, individual data points.

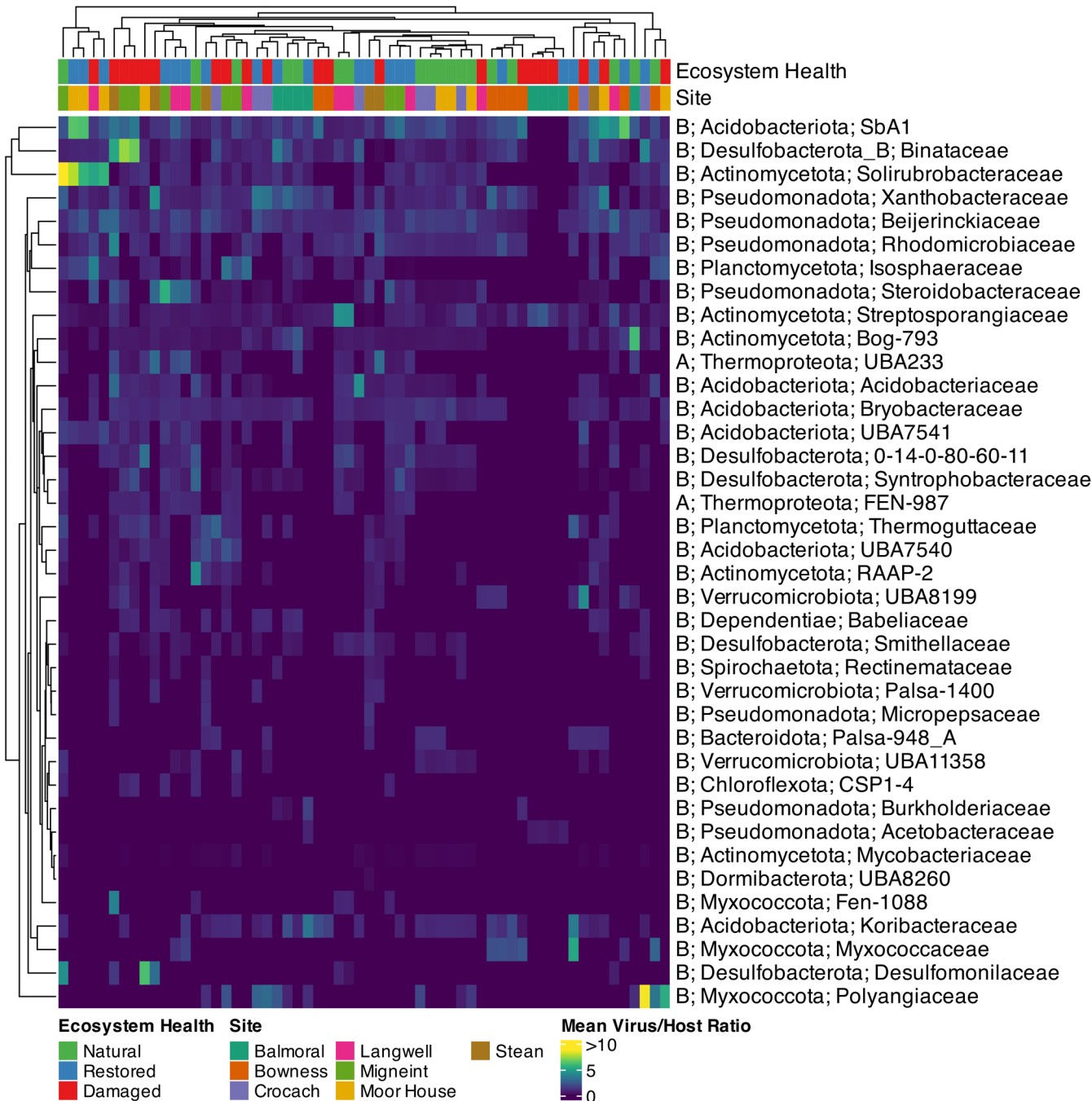

**Extended Data Fig. 6 | Average Virus/Host Abundance Ratios by Host Family.** Virus/host abundance ratios for viruses with known hosts, averaged at the family level (rows) for every metagenome assembly (columns). Families are labeled as follows: Domain (A = Archaea, B = Bacteria); Phylum; Family.

# Reporting Summary

## Statistics

For all statistical analyses, confirm that the following items are present in the figure legend, table legend, main text, or Methods section.

| n/a | Confirmed | |
|---|---|---|
| ☐ | ☒ | The exact sample size ($n$) for each experimental group/condition, given as a discrete number and unit of measurement |
| ☐ | ☒ | A statement on whether measurements were taken from distinct samples or whether the same sample was measured repeatedly |
| ☐ | ☒ | The statistical test(s) used AND whether they are one- or two-sided<br>*Only common tests should be described solely by name; describe more complex techniques in the Methods section.* |
| ☐ | ☒ | A description of all covariates tested |
| ☐ | ☒ | A description of any assumptions or corrections, such as tests of normality and adjustment for multiple comparisons |
| ☐ | ☒ | A full description of the statistical parameters including central tendency (e.g. means) or other basic estimates (e.g. regression coefficient) AND variation (e.g. standard deviation) or associated estimates of uncertainty (e.g. confidence intervals) |
| ☐ | ☒ | For null hypothesis testing, the test statistic (e.g. $F$, $t$, $r$) with confidence intervals, effect sizes, degrees of freedom and $P$ value noted<br>*Give P values as exact values whenever suitable.* |
| ☒ | ☐ | For Bayesian analysis, information on the choice of priors and Markov chain Monte Carlo settings |
| ☐ | ☒ | For hierarchical and complex designs, identification of the appropriate level for tests and full reporting of outcomes |
| ☒ | ☐ | Estimates of effect sizes (e.g. Cohen's $d$, Pearson's $r$), indicating how they were calculated |

*Our web collection on statistics for biologists contains articles on many of the points above.*

## Software and code

Policy information about availability of computer code

| Data collection | All scripts for data processing and visualization are available via GitHub at https://github.com/AnantharamanLab/UKPeatlandViruses. The software used for analyses, and their versions, are as follows: Anvi'o v8, ape v5.8 (R package), Bowtie2 v2.4.5, Bowtie2 v2.5.1, CheckM v1.2.2, ComplexUpset v1.3.3 (R package), CoverM v0.6.1, DESeq2 v1.44.0 (R package), dRep v3.5.0, factoextra v1.0.7 (R package), GeNomad v1.7.4, ggplot2 v3.5.1 (R package), ggpubr v0.6.0 (R package), GTDB-Tk v2.3.2, Illumina-utils v2.13, iPHoP v1.3.3, lme4 v1.1.35.5 (R package), mapdata v2.3.1 (R package), maps v3.4.2 (R package), MCL v14-137, MEGAHIT v1.2.9, MetaBAT 2 v2.15, METABOLIC v4, metaQUAST v5.2.0, MMseqs2 v15.6f452, MuMIn v1.48.4 (R package), pyHMMER v0.10.10 (Python package), pyrodigal-gv v0.3.1 (Python package), Python v3.10.11, R v4.4.0, SAMtools v1.17, skani v0.2.1, vegan v2.6.6.1 (R package), ViWrap v1.3.0, vRhyme v1.1.0 |
|---|---|
| Data analysis | All scripts for data processing and visualization are available via GitHub at https://github.com/AnantharamanLab/UKPeatlandViruses. The software used for analyses, and their versions, are as follows: Anvi'o v8, ape v5.8 (R package), Bowtie2 v2.4.5, Bowtie2 v2.5.1, CheckM v1.2.2, ComplexUpset v1.3.3 (R package), CoverM v0.6.1, DESeq2 v1.44.0 (R package), dRep v3.5.0, factoextra v1.0.7 (R package), GeNomad v1.7.4, ggplot2 v3.5.1 (R package), ggpubr v0.6.0 (R package), GTDB-Tk v2.3.2, Illumina-utils v2.13, iPHoP v1.3.3, lme4 v1.1.35.5 (R package), mapdata v2.3.1 (R package), maps v3.4.2 (R package), MCL v14-137, MEGAHIT v1.2.9, MetaBAT 2 v2.15, METABOLIC v4, metaQUAST v5.2.0, MMseqs2 v15.6f452, MuMIn v1.48.4 (R package), pyHMMER v0.10.10 (Python package), pyrodigal-gv v0.3.1 (Python package), Python v3.10.11, R v4.4.0, SAMtools v1.17, skani v0.2.1, vegan v2.6.6.1 (R package), ViWrap v1.3.0, vRhyme v1.1.0 |

For manuscripts utilizing custom algorithms or software that are central to the research but not yet described in published literature, software must be made available to editors and reviewers. We strongly encourage code deposition in a community repository (e.g. GitHub). See the Nature Portfolio guidelines for submitting code & software for further information.

## Data

Policy information about availability of data
All manuscripts must include a data availability statement. This statement should provide the following information, where applicable:

- Accession codes, unique identifiers, or web links for publicly available datasets
- A description of any restrictions on data availability
- For clinical datasets or third party data, please ensure that the statement adheres to our policy

All raw sequencing data are publicly available in the NCBI Short Read Archive under BioProject accession PRJNA1203648. Whole assembled metagenomic contigs as well as high-quality prokaryotic metagenome-assembled genomes are available at the NCBI WGS using the same BioProject accession. All viral metagenome-assembled genomes and prokaryotic metagenome-assembled genomes (medium and high quality) are publicly available on figshare under the DOI 10.6084/m9.figshare.28143446. Source data are provided with this paper.

## Research involving human participants, their data, or biological material

Policy information about studies with human participants or human data. See also policy information about sex, gender (identity/presentation), and sexual orientation and race, ethnicity and racism.

| | |
|---|---|
| Reporting on sex and gender | *Use the terms sex (biological attribute) and gender (shaped by social and cultural circumstances) carefully in order to avoid confusing both terms. Indicate if findings apply to only one sex or gender; describe whether sex and gender were considered in study design; whether sex and/or gender was determined based on self-reporting or assigned and methods used.*<br>*Provide in the source data disaggregated sex and gender data, where this information has been collected, and if consent has been obtained for sharing of individual-level data; provide overall numbers in this Reporting Summary. Please state if this information has not been collected.*<br>*Report sex- and gender-based analyses where performed, justify reasons for lack of sex- and gender-based analysis.* |
| Reporting on race, ethnicity, or other socially relevant groupings | *Please specify the socially constructed or socially relevant categorization variable(s) used in your manuscript and explain why they were used. Please note that such variables should not be used as proxies for other socially constructed/relevant variables (for example, race or ethnicity should not be used as a proxy for socioeconomic status).*<br>*Provide clear definitions of the relevant terms used, how they were provided (by the participants/respondents, the researchers, or third parties), and the method(s) used to classify people into the different categories (e.g. self-report, census or administrative data, social media data, etc.)*<br>*Please provide details about how you controlled for confounding variables in your analyses.* |
| Population characteristics | *Describe the covariate-relevant population characteristics of the human research participants (e.g. age, genotypic information, past and current diagnosis and treatment categories). If you filled out the behavioural & social sciences study design questions and have nothing to add here, write "See above."* |
| Recruitment | *Describe how participants were recruited. Outline any potential self-selection bias or other biases that may be present and how these are likely to impact results.* |
| Ethics oversight | *Identify the organization(s) that approved the study protocol.* |

Note that full information on the approval of the study protocol must also be provided in the manuscript.

# Field-specific reporting

Please select the one below that is the best fit for your research. If you are not sure, read the appropriate sections before making your selection.

☐ Life sciences  ☐ Behavioural & social sciences  ☒ Ecological, evolutionary & environmental sciences

For a reference copy of the document with all sections, see nature.com/documents/nr-reporting-summary-flat.pdf

# Ecological, evolutionary & environmental sciences study design

All studies must disclose on these points even when the disclosure is negative.

| | |
|---|---|
| Study description | Total metagenome sequencing of 66 soil samples collected from seven upland peatland sites across Britain. Samples represent a gradient of ecosystem health statuses: near-natural benchmark, damaged, and restored, in close proximity to each other (three replicates per ecosystem health status). |
| Research sample | Soil samples were collected from seven upland peatland sites across Britain across a gradient of climatic conditions. At each site, we sampled three areas representing near-natural benchmark, damaged, and restored ecosystem health statuses in close proximity to each other (three replicates per ecosystem health status). |
| Sampling strategy | Triplicate samples were collected at each available ecosystem health level within every site to ensure sufficient statistical power for analyses and to examine both across-site and within-site variation. |

| Data collection | Soil samples were collected using a borer in triplicate across a 5-meter transect from the top 10cm layer of soil for each site x ecosystem health level combination. |
|---|---|
| Timing and spatial scale | Samples were collected at a single time point for each site, sampling sites between May and October of 2021. Seven upland peatland sites were sampled across Scotland, England, and Wales , chosen for their internal variation in ecosystem health levels. |
| Data exclusions | Since the Langwell site had nine restored peatland replicates (instead of three, as the other sites had) representing soils with three different periods since restoration, data from only the three replicates from soils with the longest restoration period were retained to ensure consistency across sites for both community composition and downstream analyses |
| Reproducibility | Triplicate samples were collected along a 5-meter transect at each ecosystem health level within every sampling site to ensure replication and reproducibility. |
| Randomization | Samples were grouped into triplicates representing the same sample site and ecosystem health levels. Covariance that may have been introduced by this grouping was accounted for in statistical analyses. |
| Blinding | Blinding is not relevant to this study since it involved sampling of environmental soil samples and sequencing their total community metagenomes. |

Did the study involve field work?   ☒ Yes   ☐ No

## Field work, collection and transport

| Field conditions | Balmoral mean annual precipitation 1412mm mean annual temperature 5.5 C, Bowness mean annual precipitation 953mm mean annual temperature 9.6 C, Crocach mean annual precipitation 1258mm mean annual temperature 7.1 C, Langwell mean annual precipitation 1223mm mean annual temperature 7 C, Migneint mean annual precipitation 2181mm mean annual temperature 8 C, Moor House mean annual precipitation 1699mm mean annual temperature 8 C, Stean mean annual precipitation 1229mm mean annual temperature 8 C. |
|---|---|
| Location | Balmoral: 695 m elevation United Kingdom (Scotland) 56.92341 N 3.67514 W, Bowness: 66 m elevation United Kingdom (England) 54.93297 N 3.23945 W, Crocach: 189 m elevation United Kingdom (Scotland) 58.39304 N 4.00182 W, Migneint: 453 m elevation United Kingdom (Wales) 52.96932 N 3.81616 W, Moor House: 571 m elevation United Kingdom (England) 54.69457 N 2.37661 W, Stean: 530 m elevation United Kingdom (England) 54.13559 N 1.92875 W |
| Access & import/export | *Describe the efforts you have made to access habitats and to collect and import/export your samples in a responsible manner and in compliance with local, national and international laws, noting any permits that were obtained (give the name of the issuing authority, the date of issue, and any identifying information).* |
| Disturbance | *Describe any disturbance caused by the study and how it was minimized.* |

# Reporting for specific materials, systems and methods

We require information from authors about some types of materials, experimental systems and methods used in many studies. Here, indicate whether each material, system or method listed is relevant to your study. If you are not sure if a list item applies to your research, read the appropriate section before selecting a response.

## Materials & experimental systems

| n/a | Involved in the study |
|---|---|
| ☒ ☐ | Antibodies |
| ☒ ☐ | Eukaryotic cell lines |
| ☒ ☐ | Palaeontology and archaeology |
| ☒ ☐ | Animals and other organisms |
| ☒ ☐ | Clinical data |
| ☒ ☐ | Dual use research of concern |
| ☒ ☐ | Plants |

## Methods

| n/a | Involved in the study |
|---|---|
| ☒ ☐ | ChIP-seq |
| ☒ ☐ | Flow cytometry |
| ☒ ☐ | MRI-based neuroimaging |

## Plants

Seed stocks

*Report on the source of all seed stocks or other plant material used. If applicable, state the seed stock centre and catalogue number. If plant specimens were collected from the field, describe the collection location, date and sampling procedures.*

Novel plant genotypes

*Describe the methods by which all novel plant genotypes were produced. This includes those generated by transgenic approaches, gene editing, chemical/radiation-based mutagenesis and hybridization. For transgenic lines, describe the transformation method, the number of independent lines analyzed and the generation upon which experiments were performed. For gene-edited lines, describe the editor used, the endogenous sequence targeted for editing, the targeting guide RNA sequence (if applicable) and how the editor was applied.*

Authentication

*Describe any authentication procedures for each seed stock used or novel genotype generated. Describe any experiments used to assess the effect of a mutation and, where applicable, how potential secondary effects (e.g. second site T-DNA insertions, mosiacism, off-target gene editing) were examined.*

