## [Peer Review File · Nature Microbiology]

Ecosystem Health Shapes Viral Ecology in Peatland Soils

Corresponding Author: Dr Karthik Anantharaman

Version 0:

Decision Letter:

20th February 2025

Dear Karthik,

Thank you for your patience while your manuscript "Ecosystem Health Shapes Viral Ecology in Peatland Soils" was under peer-review at Nature Microbiology. It has now been seen by 3 referees, whose expertise and comments you will find at the end of this email. Although they find your work to be timely and of potential interest, they have raised a number of concerns that will need to be addressed before we can consider publication of the work in Nature Microbiology.

In particular, the main issue here is that the claims about peatland restoration aren't really connected to the data—from an editorial perspective this is the major selling point for us, so this will need to be made more robust in a revision, both experimentally (as the reviewers point out, this needs stronger mechanistic evidence, better connection to broader ecological principles and environmental data, and additional functional validation) and editorially (the semantics of how the peatland characteristics are discussed and how this actually ties into viral dynamics—how do you suppose restoration experts would actually use these findings?). There were also some additional technical issues pointed out too, in particular related to the assemblies.

Should further experimental data allow you to address these criticisms, we would be happy to look at a revised manuscript.

Please include a data availability statement as a separate section after Methods but before references, under the heading "Data Availability". This section should inform readers about the availability of the data used to support the conclusions of your study. This information includes accession codes to public repositories (data banks for protein, DNA or RNA sequences, microarray, proteomics data etc...), references to source data published alongside the paper, unique identifiers such as URLs to data repository entries, or data set DOIs, and any other statement about data availability. At a minimum, you should include the following statement: "The data that support the findings of this study are available from the corresponding author upon request", mentioning any restrictions on availability. If DOIs are provided, we also strongly encourage including these in the Reference list (authors, title, publisher (repository name), identifier, year). For more guidance on how to write this section please see: <http://www.nature.com/authors/policies/data/data-availability-statements-data-citations.pdf>

* If you have not done so already we suggest that you begin to revise your manuscript so that it conforms to our Article format instructions at <http://www.nature.com/nmicrobiol/info/final-submission>. Refer also to any guidelines provided in this letter.

* Include a revised version of any required reporting checklist. It will be available to referees (and, potentially, statisticians) to aid

in their evaluation if the manuscript goes back for peer review. A revised checklist is essential for re-review of the paper.

EXTENDED DATA FIGURES

Link Redacted

Note: This url links to your confidential homepage and associated information about manuscripts you may have submitted or be reviewing for us. If you wish to forward this e-mail to co-authors, please delete this link to your homepage first.

Nature Microbiology is committed to improving transparency in authorship. As part of our efforts in this direction, we are now requesting that all authors identified as 'corresponding author' on published papers create and link their Open Researcher and Contributor Identifier (ORCID) with their account on the Manuscript Tracking System (MTS), prior to acceptance. This applies to primary research papers only. ORCID helps the scientific community achieve unambiguous attribution of all scholarly contributions. You can create and link your ORCID from the home page of the MTS by clicking on 'Modify my Springer Nature account'. For more information please visit www.springernature.com/orcid.

If you wish to submit a suitably revised manuscript we would hope to receive it within 6 months. If you cannot send it within this time, please let us know. We will be happy to consider your revision, even if a similar study has been accepted for publication at Nature Microbiology or published elsewhere (up to a maximum of 6 months).

Yours sincerely,

[Signature redacted]

Reviewer Expertise:

- Referee #1: peatland biogeochem, metagenomics, soil microbiology
- Referee #2: metagenomics, soil viruses, microbial ecology
- Referee #3: peatland biogeochem, peatland restoration, soil microbiology

Reviewer Comments:

Reviewer #1 (Remarks to the Author):

This manuscript by Kosmopoulos et al. describes relationships between viral and host microbial communities across peatlands, with a focus on differences by peatland "health". The authors had 66 metagenomes from 7 different peatlands in the UK, including natural, restored, and damaged locations. They coassembled these metagenomes and recovered a final dataset of 2,281 vMAGs and 935 host MAGs. They then used the abundances of vMAGs and host MAGs to assess community composition across locations and peatland type. The main finding was that viral populations and functional potential varied by ecosystem health. I think this manuscript is clearly written, the figures are helpful, and the metagenomic processing well done. I appreciate all of the supplemental data and the github is helpful. My main concern is about the use of the health categories and the lack of data in support of them, given all conclusions in the manuscript derive from these categories. My specific comments that I would

like to see addressed are:

1. I struggle with the wording that these peatlands are different based on ecosystem health. I understand that low disturbance peatlands and damaged peatlands are different in their hydrology and geochemistry and gas emissions, and I understand what the authors are trying to capture by saying they vary in health. However, health implies performance, and there is no data provided to say, for example, Balmoral Natural peat is healthier than Crocach Damaged peat, or that Crocach Damaged peat is of equal health with Langwell Damaged peat. Would the authors please consider and justify their use of the term health?
2. Related to the idea above, the use of these categories would be greatly supported by environmental data. The authors speculate on some environmental differences in their discussion of functions, but it would be great to have measurements to say natural peat has similar hydrology/geochemistry and damaged peat has similar hydrology/geochemistry.
3. The authors mention that the 6-years since restoration samples were selected for Langwell (lines 453-457), but how long since restoration were the other sites? Was it the same for all restored samples? Similarly, were all drained sites drained at similar times?
4. Line 333-334: how close is "close proximity"-meters? Kilometers? Is this standardized across locations? Without knowing this, I am left to wonder if differences in Fig. 1C, for example, are really from ecosystem health status or if they are just several km apart.
5. What depth did the peat samples come from- was it the same across the 66 samples? How did sampling depth compare to the water table for each site?
6. Line 342- is this missing a reference? Where was this previously described?
7. Please ensure the metagenomes are publicly available. I tried to verify using PRJNA1203648, but NCBI did not return any public results for this.

Reviewer #1 (Remarks on code availability):

I reviewed the github and it appeared thorough for completing the analyses and reproducing the figures.

Reviewer #2 (Remarks to the Author):

Kosmopoulos and colleagues report on the biogeography, endemism, population dynamics, structure, and function of DNA viral communities, in relation to their identified hosts, across a peatland soil ecosystem health gradient critical for understanding planetary carbon stabilization and loss processes. The biological entities (viruses), the ecosystem (peatland soils), and the research findings (dynamics across natural, damaged, and restored systems) are all important and relevant to the collective scientific efforts to understand how our planet will respond to global change. The manuscript is well-written, the data and code are well-organized, and the results are well-presented. I have only a few suggestions for the authors to further improve their manuscript.

Main issue: The discussions around 'piggyback-the-winner' dynamics need to be greatly qualified by explicitly stating the technical limitations that can wildly impact the 'piggyback-the-winner' conclusion. Soil environments are very challenging when it comes to studying viruses. Are the viruses captured in the microbial metagenomes biased towards those protected inside their host cells? Do humics occlude seeing the genetic material of the majority of free viral particles? Regardless of the virus identification, lifestyle prediction, and host-linkage tools used in this work, do bioinformatic tools comprehensively capture the full diversity of environmental viruses, their definitive lifestyle, and their full host range? Does host phylum-level aggregation of abundances reflect relevant species-level ecological interactions?

Minor issues:

- Please review all the places where you cite main/supplementary figures and tables as some of these need to be updated. For example, Line 245 in the main text should call Table S3, correct? Similarly, the text (and figure/table citations) on lines 45-46 in the supplementary text should be updated to reflect the host results, rather than the virus results.
- Figure 5: even though the authors did a great job correcting for multiple testing throughout the manuscript, the linear regression analyses in figure 5A do not seem to be corrected for multiple statistical testing.

Reviewer #2 (Remarks on code availability):

The authors provided a link to a GitHub repository that hosts the scripts and input tables necessary to replicate the results shown in their manuscript. The repository is impressively well-organized and both the basic bioinformatic and special figure-generating scripts were included as workflows and notebooks (such as R markdown notebooks). These notebooks display the R code used for each main and supplementary figure alongside the output figures, tables, and results of various statistical tests. The authors should make sure that the data hosted in the NCBI SRA (BioProject accession PRJNA1203648) and on figshare (<https://doi.org/10.6084/m9.figshare.28143446>) are live to the public by the time of the manuscript publication as I currently cannot access these two locations (I, however, do have access to the MAG and vMAG sequences via the private link that the authors shared during their manuscript submission).

Reviewer #3 (Remarks to the Author):

Synopsis, major strengths and weaknesses: This study presents a comprehensive investigation of viral communities and virus-host dynamics across UK peatland sites with different ecosystem health states. Through metagenomic analysis, the authors characterize viral communities, their hosts, and their functions across natural, damaged, and restored peatland conditions. Their findings demonstrate that both geography and ecosystem health drive viral community structure, with viruses showing ecosystem health-specific endemism and varying virus-host dynamics across different health states.

The main strength of the paper lies in its comprehensive viral characterization across seven peatlands in different states (natural, damaged, and restored), which adds valuable new information to the literature about virus-host dynamics in these critical ecosystems. The manuscript is well-written throughout, with clear and informative figures that effectively communicate complex relationships between viruses and their hosts.

However, several significant weaknesses warrant attention. The study struggles to connect its viral findings to practical restoration applications, and some key conclusions, particularly regarding methanogens, are based on limited data (only 1 virus and 2 hosts). There are also inconsistencies in the reporting of endemic viruses (54%) versus abundance changes (94%), and the temporal sampling/sampling details are limited.

Summary: This manuscript represents a strong contribution to viral ecology and ecosystem restoration but requires significant revision before publication. While the core findings are novel and important, particularly in linking viral dynamics to ecosystem health status, the manuscript needs stronger mechanistic evidence, better connection to broader ecological principles, additional functional validation, and improved theoretical framing. With these enhancements, particularly focusing on mechanistic understanding and practical restoration applications, the study could make a compelling contribution to the field.

Please see below more specific details:

SECTION SPECIFIC COMMENTS:

Abstract specific comments: The abstract clearly presents the study's main findings, but the logical flow could be improved. For example, the abstract starts by stating viruses shape microbiomes and impact ecosystems, then jumps to peatlands degrading. While both points are important, it doesn't clearly establish why studying viruses specifically in peatlands is crucial. A stronger connection could be made between these opening statements. Additionally, while the study examines viral communities across different peatland health statuses, the findings don't clearly tie back to the initial problem of peatland degradation. While they found that viruses respond to ecosystem health and show specific patterns, the abstract doesn't explicitly explain how this knowledge helps with restoration efforts. The conclusion states that viruses are pivotal for restoration and climate mitigation, but the abstract doesn't fully support this claim. Making a clearer connection between the findings and their implications for restoration would strengthen the abstract.

Introduction specific comments: The introduction effectively builds from broad context (peatlands' global importance) to specific focus (viral communities in restoration) but has several key areas that could be strengthened for greater impact. While it successfully establishes peatlands' significance for carbon storage and the threat of degradation, the connection between viral dynamics and restoration success needs more development. For example, the middle part talks about viruses and their role in ecosystems. It brings up some good examples from other peatland studies, showing how viruses affect carbon cycling and respond to water levels. But these examples feel a bit disconnected - like a list of facts rather than a story about why viruses matter for restoration. The biggest weakness is in explaining why we should care about viruses when trying to restore peatlands. The introduction tells us no one has studied viruses across healthy, damaged, and restored peatlands before. But it doesn't clearly tell us why these matters. What could we do differently in restoration if we understood viral patterns better? How might this knowledge help us restore peatlands more successfully? These questions aren't answered clearly enough. If space permits, I think it's also important to add some predictions to the end of the introduction. For example, what patterns would you expect to find or why those patterns would matter for restoration work? Adding these predictions would help readers understand why this research is useful, not just interesting. I don't think the paper had any predictions or hypothesis.

Results specific comments: I have some questions about the metagenome assembly results in your study. You mention that co-assembling 66 samples yielded 22 high-quality metagenomes, which seems like a relatively low success rate of about 33% compared to other peatland studies. Could you clarify how much of this is related to particularly stringent quality criteria, wanting 90% completeness and less than 10% contamination? Did you try different cutoffs to see if you could rescue more samples while keeping good quality? It would be helpful to understand whether this lower recovery rate was due to quality criteria, or technical challenges with specific sites, or other factors. I did find more information about this in the SI files, however it left me more confused. In the main text, you say co-assembling 66 samples yielded 22 high-quality metagenomes, but the SI shows these were actually co-assemblies that led to 935 medium and high-quality MAGs. Could you clarify what you mean by '22 high-quality metagenomes'? Are these the same as your 22 co-assemblies? Also, could you explain why you chose to make 22 co-assemblies from 66 samples - what was the logic behind this grouping? Did I miss anything? Or misunderstood your approach? Did you notice any patterns in which samples yielded high-quality assemblies - for instance, were natural sites more successful than disturbed ones? Additionally, could you discuss how this relatively low number of high-quality MAGs might have influenced your downstream analyses and conclusions about viral-host relationships? Can you provide more details on sampling? What depth were the samples collected from? Were they surface or deeper samples? How did you ensure valid comparisons across different sites? What was the water table depth, and how was it adjusted across sites? Also, when were the samples collected, and how might seasonality affect the results? Depth could have a strong impact on the findings.

I am also a bit confused about how you compared their viruses to existing databases. You say you found more matches with other databases than within your own study, but it's not really clear what this means? It would really help to know how these patterns compare to other types of soil. Are peatland viruses more widespread than viruses in, say, forest soils? Also, are certain types of viruses better at spreading around than others? Can you expand a little bit more on Figure 2 in the text.

Lines 115-134 Earlier the authors said 54% of viruses were 'endemic' to specific ecosystem states, but here they are showing that 94% of viruses change in abundance across states. This seems contradictory - are these viruses present but rare in some states and abundant in others?

lines 136-153. Could you provide more context about the virus-host abundance discordance - what mechanisms might explain

cases where viral populations increase while host populations remain stable?
lines 154-187 The methanogen virus conclusions are based on very small numbers (1 virus, 2 hosts) - how reliable are these patterns? Could you provide power analyses or additional validation? Also, for the reported changes in fermentation (-22%) and carbohydrate degradation (-15%), were these changes statistically significant?"
Additionally, while you show viral protein adaptation across ecosystem states, how might these adaptations influence restoration success?

lines 222-256) The finding of only 13% lysogenic viruses seems potentially low - how does this compare to other soil environments? Could technical limitations in identifying lysogenic viruses affect this percentage? You show opposing trends in absolute versus relative abundance of lysogenic viruses in damaged soils - what are the implications for restoration practices? Could you explain how you selected the eight key metabolic functions for analysis, and whether other important functions might have been missed.

Discussion section: While the results section provides a lot insights and potentially new findings, unfortunately the discussion falls short in several important ways. While the authors found interesting patterns in viral communities across peatland sites, they don't clearly explain how these findings could help restore peatlands. This is my major critique of the paper. The authors mentioned multiple times that viruses are 'pivotal' and 'critical' for restoration, but never tell us what this means in practice. Another issue is that the authors talked about viruses controlling methanogens -I found this very exciting with a lot of implications however this big claim is based on just one virus and two host genomes. How important is this virus then? (maybe expand on this on LINES 284-287.)

The authors also skip over some big questions from their results. They never explain why they found different numbers for endemic viruses (54%) versus viruses that change in abundance (94%).

It will be good if the authors expand in the main text or in supplementary on how the methods of choice, and while might undercount viruses, how this could affect their conclusions. You only looked at 22 MAGs, how representative is that of real dynamics?> were these 22 MAGs representative of the full community?

The ending feels weak. Instead of offering specific next steps, the authors give vague suggestions like 'future research should explore mechanisms'. They don't describe any experiments that could help prove their ideas about viruses in restoration. Most importantly, they don't explain how restoration experts could use these findings to improve their work. For a paper focused on peatland restoration, this is a significant gap. If the goal of the paper is simply to characterize viruses, I think that's fine. However, the authors will need to remove the claim that this will help with restoration, as it doesn't follow from the current findings.

The broader ecological context could also be better developed to demonstrate impact beyond peatland systems. The ecological context could be enhanced by better connecting findings to general ecological theory and expanding implications beyond peatland systems. The presentation could also be improved by emphasizing novelty more clearly in the introduction and strengthening connections to broader ecological principles and climate change implications

One minor thing: I couldn't access this link: <https://doi.org/10.6084/m9.figshare.28143446>. (DOI Not Found). I am assuming it is not public yet

Reviewer #3 (Remarks on code availability):

the DOI link was not working. The GitHub link worked but I didn't check the code.

Version 1:

Decision Letter:

Our ref: NMICROBIOL-25010039A

22nd September 2025

Dear Karthik,

Thank you for submitting your revised manuscript "Ecosystem Health Shapes Viral Ecology in Peatland Soils" (NMICROBIOL-25010039A). It has now been seen by the original referees and their comments are below. The reviewers find that the paper has improved in revision, and therefore we'll be happy in principle to publish it in Nature Microbiology, pending minor revisions to satisfy the referees' final requests and to comply with our editorial and formatting guidelines.

Thank you again for your interest in Nature Microbiology Please do not hesitate to contact me if you have any questions.

Sincerely,

[Signature redacted]

Reviewer #1 (Remarks to the Author):

Thank you to the authors for considering and addressing my comments. I appreciate the inclusion of the environmental data and the health index, and feel they provided useful context for the site categories. All my concerns have been addressed.

Reviewer #1 (Remarks on code availability):

The github looks comprehensive. I understand the reviewers are keeping the data private before it is public, but the github links to Bioproject and figshare links will need to be made public.

Reviewer #2 (Remarks to the Author):

No further comments. Thank you for addressing my previous questions and requests.

Reviewer #2 (Remarks on code availability):

No further comments.

Reviewer #3 (Remarks to the Author):

I appreciate that the authors responded thoroughly to my comments and the other reviewers' feedback and made substantial improvements to the manuscript. I especially liked the new environmental data and the ecosystem health index. It makes a lot stronger case for their health categories. Also, thanks for clearing up the confusion between the 54% endemic viruses versus 94% differentially abundant viruses, and the MAGs versus co-assemblies.

Version 2:

Decision Letter:

20th October 2025

Dear Karthik,

I am pleased to accept your Article "Ecosystem Health Shapes Viral Ecology in Peatland Soils" for publication in Nature Microbiology. Thank you for having chosen to submit your work to us and many congratulations.

You may wish to make your media relations office aware of your accepted publication, in case they consider it appropriate to organize some internal or external publicity. Once your paper has been scheduled you will receive an email confirming the publication details. This is normally 3-4 working days in advance of publication. If you need additional notice of the date and time of publication, please let the production team know when you receive the proof of your article to ensure there is sufficient time to coordinate. Further information on our embargo policies can be found here:

<https://www.nature.com/authors/policies/embargo.html>

Due to the importance of these deadlines, we ask you please us know now whether you will be difficult to contact over the next month. If this is the case, we ask you provide us with the contact information (email, phone and fax) of someone who will be able

to check the proofs on your behalf, and who will be available to address any last-minute problems.

Authors may need to take specific actions to achieve compliance with funder and institutional open access mandates. If your research is supported by a funder that requires immediate open access (e.g. according to [a Plan S principles](https://www.springernature.com/gp/open-science/plan-s-compliance) or the [NIH public access policy](https://www.springernature.com/gp/open-science/us-federal-agency-compliance)) then you should select the gold OA route, and we will direct you to the compliant route where possible. Because authors warrant under our subscription licensing terms that they haven't committed to licensing any version of their article under a licence inconsistent with the terms of our agreement – including the applicable embargo period – publication under the subscription model isn't suitable for authors whose funders require no embargo.

With kind regards,

[Signature redacted]

P.S. Click on the following link if you would like to recommend Nature Microbiology to your librarian <http://www.nature.com/subscriptions/recommend.html#forms>

** Visit the Springer Nature Editorial and Publishing website at http://editorial-jobs.springernature.com?utm_source=ejP_NMicro_email&utm_medium=ejP_NMicro_email&utm_campaign=ejP_NMicro for more information about our career opportunities. If you have any questions please click [here](mailto:editorial.publishing.jobs@springernature.com).**

Response to Referees

We thank the editor and reviewers for their thorough assessment and valuable feedback, which significantly improved our manuscript. In response to the reviewers' comments, we have made several substantial revisions:

1. We have restructured our abstract and introduction to more clearly link soil viral communities to peatland restoration.
2. We have clarified and justified our use of "ecosystem health" categories by adding environmental data and analyses (see Figure 1A, Figure 1D-E, Extended Data Figure 1, and Figure 5C) to demonstrate meaningful distinctions between natural, damaged, and restored peatlands using both a categorical and a continuous, data-driven measurement of ecosystem health.
3. Additionally, we expanded the discussion section to explicitly address the practical implications of our findings for peatland restoration, highlighting how knowledge of viral and microbial diversity can inform both restoration design and monitoring.
4. Methodological clarifications were also included to clearly distinguish between metagenome co-assemblies and metagenome-assembled genomes (MAGs), resolving previous confusion about representativeness and quality criteria.
5. We carefully qualified our conclusions drawn from limited data, particularly regarding methanogen-infecting viruses, and clarified the distinction between endemicity and differential abundance analyses.
6. Lastly, we enhanced the manuscript's broader ecological context by explicitly linking our findings to general microbial ecology and restoration implications.

We hope these revisions, detailed point-by-point below, comprehensively address the reviewers' concerns and clearly communicate the broader relevance and applicability of our findings.

Reviewer Expertise:

Referee #1: peatland biogeochem, metagenomics, soil microbiology

Referee #2: metagenomics, soil viruses, microbial ecology

Referee #3: peatland biogeochem, peatland restoration, soil microbiology

Reviewer Comments:

Reviewer #1 (Remarks to the Author):

This manuscript by Kosmopoulos et al. describes relationships between viral and host

microbial communities across peatlands, with a focus on differences by peatland “health”. The authors had 66 metagenomes from 7 different peatlands in the UK, including natural, restored, and damaged locations. They coassembled these metagenomes and recovered a final dataset of 2,281 vMAGs and 935 host MAGs. They then used the abundances of vMAGs and host MAGs to assess community composition across locations and peatland type. The main finding was that viral populations and functional potential varied by ecosystem health.

I think this manuscript is clearly written, the figures are helpful, and the metagenomic processing well done. I appreciate all of the supplemental data and the github is helpful. My main concern is about the use of the health categories and the lack of data in support of them, given all conclusions in the manuscript derive from these categories.

We understand the importance of clearly defining and supporting our use of “ecosystem health” categories (natural, restored, damaged). In the revised manuscript, we have explicitly justified this terminology and provided additional data to support the distinctions among these categories.

First, we have introduced new environmental measurements for each site (soil pH, moisture content, conductivity, oxygen concentration, total carbon, total nitrogen) and summarized them with a principal components analysis (PCA), outlined in Extended Data Figure 1. This analysis shows that samples cluster according to ecosystem health, indicating that “natural,” “restored,” and “damaged” peatlands indeed differ in their environmental conditions.

Second, we have also calculated an “ecosystem health index” for each sample, incorporating peat chemistry, dissolved oxygen, moisture levels, and vegetation composition (details provided in Pallier et al. 2025 bioRxiv). This index provides a holistic, continuous measure of peatland ecosystem health that effectively reflects variation across ecosystem health statuses and sites. We now show the variation in this health index across samples and ecosystem health statuses, allowing direct comparisons between sample sites. We also linked this health index to viral community composition in Figures 1D-E and to lysogenic virus abundance in Figure 5C, providing more evidence that ecosystem health strongly structures viral communities and their functions both within and between sites.

By including the new environmental data (Figure 1D-E, Figure 5C, Extended Data Figure 1), we hope to reassure the reviewer that the “ecosystem health” categories are grounded in measurable differences.

My specific comments that I would like to see addressed are:

1. I struggle with the wording that these peatlands are different based on ecosystem health. I understand that low disturbance peatlands and damaged peatlands are different in their hydrology and geochemistry and gas emissions, and I understand what the authors are trying to capture by saying they vary in health. However, health implies

performance, and there is no data provided to say, for example, Balmoral Natural peat is healthier than Crocach Damaged peat, or that Crocach Damaged peat is of equal health with Langwell Damaged peat. Would the authors please consider and justify their use of the term health?

We appreciate this concern and have revised the manuscript to better justify the term “ecosystem health.” In our study, ecosystem health is defined operationally to distinguish peatlands by their disturbance and restoration status (i.e., near-natural baseline, damaged by drainage or erosion, and restored by rewetting through drain blocking). To support these distinctions with data, we have included new environmental measurements as well as a continuous ecosystem health index reflecting the ecological condition of each site (see above for details). This index enables direct, quantitative comparisons across peatland sites and accounts for cases where the natural area in a site may exhibit worse environmental conditions than the damaged area in another (Figure 1A). It also provides a gradient along which we relate viral diversity to underlying environmental variation (Figures 1D-E, 5C). While we do not measure productivity or performance directly, our index captures key features that are relevant to the ecological integrity of peatlands, in line with established definitions of ecosystem health as the combination of structural condition, functional capacity, and resilience (Worral et al., 2025 *Sci. Total Environ.*; van Bruggen et al., 2019 *Sci. Total Environ.*).

2. Related to the idea above, the use of these categories would be greatly supported by environmental data. The authors speculate on some environmental differences in their discussion of functions, but it would be great to have measurements to say natural peat has similar hydrology/geochemistry and damaged peat has similar hydrology/geochemistry.

We fully agree and have now included environmental data for our soil samples. In the revised Extended Data Figure 1, we present a PCA of measured soil parameters (soil pH, moisture content, conductivity, oxygen concentration, total carbon, total nitrogen) across all sites and ecosystem health statuses, as well as for each site individually. The PCAs show clustering of samples by ecosystem health when considering all sites, together (revised Extended Data Figure 1A) and each site, separately (Extended Data Figure 1B), indicating that these categories capture real environmental differences. We also added two panels to Figure 1 (D and E) that relate viral diversity to ecosystem health index, which incorporates peat chemistry, dissolved oxygen, moisture levels, and vegetation composition. Mapping ecosystem health index onto our viral community PCoA in Figure 1D, and performing a linear regression of the index against PCoA axis 1 in Figure 1E, shows more support for structuring of viral community by ecosystem health. By providing these measurements, we hope we address the reviewer’s request and substantiate the differences among natural, damaged, and restored peatlands beyond speculation.

3. The authors mention that the 6-years since restoration samples were selected for Langwell (lines 453-457), but how long since restoration were the other sites? Was it

the same for all restored samples? Similarly, were all drained sites drained at similar times?

We apologize for not including these details in the original submission. We have now added additional information in the Methods and in the main text to clarify the timing of drainage and restoration for each site. In brief, while all restored sites were rewetted within the last ~10 years, Langwell's restored area had been rewetted 6 years before sampling. For the damaged (drained) sites, the drainage generally occurred decades ago (exact dates are often not documented, but they likely span several decades in the past). The age of restoration varied across sites, it was 3 years for Balmoral, 6 years for Langwell and 9-12 years for the other sites. We have also cited a companion study (Pallier et al., 2025 bioRxiv) for further details on the site histories and sampling design (Methods, Line 437 in the revised manuscript).

4. Line 333-334: how close is "close proximity"-meters? Kilometers? Is this standardized across locations? Without knowing this, I am left to wonder if differences in Fig. 1C, for example, are really from ecosystem health status or if they are just several km apart.

We have clarified the sampling scheme in the Methods to define "locally adjacent" sampling areas more precisely. At each site, the plots representing the different ecosystem health statuses were located in the same watershed, ranging from tens of meters to few kilometers (see coordinates now available in revised Supplementary Table S1). In practical terms, this means that for a given site, the natural, damaged, and restored sampling areas were within the same contiguous peatland which minimizes variation in underlying geology and local climate. The use of an ecosystem health index also considers this variation within each site. We now state in the Methods (Lines 431-435): "At each site, we sampled three areas with different ecosystem health statuses (EHS): a near-natural reference (natural), damaged by drainage or erosion (damaged), and restored by rewetting through drain blocking (restored). Three replicates were sampled per EHS that were locally adjacent to minimize the impact of underlying geology and climatic conditions."

5. What depth did the peat samples come from- was it the same across the 66 samples? How did sampling depth compare to the water table for each site?

All peat soil samples in our study spanned the top 50 cm of the peat profile, with metagenomic DNA extractions performed on the upper 10 cm of the profile, and this was consistent across all 66 samples (3 replicates × 3 statuses × 7 sites + extras). We have added this detail to the Methods (Lines 443-444), noting that we homogenized the 0-10 cm layer for DNA extraction. The mean annual water table ranged from 3-20 cm (we have data for 4 of the 7 sites). The water table is deeper in the damaged areas compared to the natural and restored. For our metagenomics analysis, we chose to sample the top 10 cm to study the most active and dynamic zone of the peat column.

6. Line 342- is this missing a reference? Where was this previously described?

We apologize for initially missing this reference. We have now added the appropriate reference to the companion methodology description. Specifically, we cite Pallier et al. (2025) for details on sampling methodology, DNA extraction, and sequencing protocols.

7. Please ensure the metagenomes are publicly available. I tried to verify using PRJNA1203648, but NCBI did not return any public results for this.

We apologize for the inconvenience, we have provided the reviewer link to the SRA data on NCBI with our resubmission (<https://dataview.ncbi.nlm.nih.gov/object/PRJNA1203648?reviewer=p9jgmg4sfu73mlka594qk6i5le>). While the assembled metagenomic contigs and MAGs remain embargoed until publication, the host MAGs and vMAGs are on FigShare and can be accessed by the reviewer with the provided reviewer link in the initial submission (<https://figshare.com/s/0a3e06682400a1523c21>).

Reviewer #1 (Remarks on code availability):

I reviewed the github and it appeared thorough for completing the analyses and reproducing the figures.

Thank you for your positive assessment.

Reviewer #2 (Remarks to the Author):

Kosmopoulos and colleagues report on the biogeography, endemism, population dynamics, structure, and function of DNA viral communities, in relation to their identified hosts, across a peatland soil ecosystem health gradient critical for understanding planetary carbon stabilization and loss processes. The biological entities (viruses), the ecosystem (peatland soils), and the research findings (dynamics across natural, damaged, and restored systems) are all important and relevant to the collective scientific efforts to understand how our planet will respond to global change. The manuscript is well-written, the data and code are well-organized, and the results are well-presented. I have only a few suggestions for the authors to further improve their manuscript.

We thank Reviewer #2 for the positive assessment of our manuscript. We are pleased that the importance of the study and the organization of data/code were noted. We address the suggestions point-by-point below.

Main issue: The discussions around 'piggyback-the-winner' dynamics need to be greatly qualified by explicitly stating the technical limitations that can wildly impact the 'piggyback-the-winner' conclusion. Soil environments are very challenging when it comes to studying viruses. Are the viruses captured in the microbial metagenomes biased towards those protected inside their host cells? Do humics occlude seeing the genetic material of the majority of free viral particles? Regardless of the virus identification, lifestyle prediction, and host-linkage tools used in this work, do

bioinformatic tools comprehensively capture the full diversity of environmental viruses, their definitive lifestyle, and their full host range? Does host phylum-level aggregation of abundances reflect relevant species-level ecological interactions?

This is an excellent point. We have now added a dedicated discussion of methodological limitations in the revised manuscript to qualify our interpretations of ‘piggyback-the-winner’ dynamics and other findings. We explicitly acknowledge that soil metagenomic approaches may under-sample certain viruses, and that our results must be viewed in light of these constraints. Specifically, we now state (Discussion, Lines 388-400):

“However, our interpretations of viral lifestyle require cautious consideration of methodological limitations inherent to soil viral ecology. Microbial metagenomic approaches often bias towards intracellular or host-associated viruses, potentially missing substantial fractions of the viral community that exist as free particles (Santos-Medellin et al., 2021 ISME J.; Kosmopoulos et al. 2024 Microbiome). Humic compounds and soil complexity further complicate the recovery of viral genomic material, limiting comprehensive characterization (Wnuk et al., 2020 PeerJ; Trubl et al., 2020 Soil Systems; Anesio et al., 2004 Appl. Environ. Microbiol.). Additionally, the aggregation of abundance data at the host phylum level could obscure critical, species-specific interactions. Despite these constraints, current can recover most environmental viruses from metagenomes (Hegarty et al., 2024 mSystems; Camargo et al., 2024 Nat. Biotech.) and increasingly offer reliable host range predictions (Roux et al., 2023 PLOS Biol.). While annotation- and alignment-based methods for detecting viruses capable of lysogeny likely suffer from low recall, their high precision supports confident identification of lysogenic viruses (Hockenberry & Wilke, 2021 PeerJ). Our strong correlations between predicted virus-host pairs emphasize non-lytic virus-host coexistence as a significant replication strategy across varying soil conditions.”

We hope that these additions address the reviewer’s questions by citing relevant literature and pointing out that our observation of piggyback-the-winner is subject to detection biases. By discussing these limitations openly, we temper the interpretation of piggyback-the-winner and ensure that our conclusions are presented in a properly qualified manner. Even with these caveats, we believe that our core conclusions (e.g., shifts in viral lifestyle frequencies across health statuses) remain supported.

Minor issues:

- Please review all the places where you cite main/supplementary figures and tables as some of these need to be updated. For example, Line 245 in the main text should call Table S3, correct? Similarly, the text (and figure/table citations) on lines 45-46 in the supplementary text should be updated to reflect the host results, rather than the virus results.

We have carefully reviewed and corrected all figure and table citation numbers in both the main text and Supplementary Information.

- Figure 5: even though the authors did a great job correcting for multiple testing throughout the manuscript, the linear regression analyses in figure 5A do not seem to be corrected for multiple statistical testing.

Thank you for bringing this to our attention. In the original submission, the *P*-values for the regression slopes in Figure 5A were indeed unadjusted. We have now recomputed those regression analyses with a Benjamini-Hochberg (BH) correction for multiple comparisons. In the revised Figure 5A, all *P*-values for slope differences are reported as BH-adjusted *P*. We also updated the Results accordingly (lines 287-289) to state that these regressions were evaluated with adjusted *P*-values. This did not change our interpretation: every regression that was significant before (unadjusted $P < 0.05$) remained significant after adjustment, and those that were non-significant remained non-significant.

Reviewer #2 (Remarks on code availability):

The authors provided a link to a GitHub repository that hosts the scripts and input tables necessary to replicate the results shown in their manuscript. The repository is impressively well-organized and both the basic bioinformatic and special figure-generating scripts were included as workflows and notebooks (such as R markdown notebooks). These notebooks display the R code used for each main and supplementary figure alongside the output figures, tables, and results of various statistical tests.

Thank you for the positive feedback on our code repository.

The authors should make sure that the data hosted in the NCBI SRA (BioProject accession PRJNA1203648) and on figshare (<https://doi.org/10.6084/m9.figshare.28143446>) are live to the public by the time of the manuscript publication as I currently cannot access these two locations (I, however, do have access to the MAG and vMAG sequences via the private link that the authors shared during their manuscript submission).

Please see our response to reviewer #1 comment 7, above.

Reviewer #3 (Remarks to the Author):

Synopsis, major strengths and weaknesses: This study presents a comprehensive investigation of viral communities and virus-host dynamics across UK peatland sites with different ecosystem health states. Through metagenomic analysis, the authors characterize viral communities, their hosts, and their functions across natural, damaged, and restored peatland conditions. Their findings demonstrate that both geography and ecosystem health drive viral community structure, with viruses showing ecosystem health-specific endemism and varying virus-host dynamics across different health states.

The main strength of the paper lies in its comprehensive viral characterization across seven peatlands in different states (natural, damaged, and restored), which adds valuable new information to the literature about virus-host dynamics in these critical ecosystems. The manuscript is well-written throughout, with clear and informative figures that effectively communicate complex relationships between viruses and their hosts.

However, several significant weaknesses warrant attention. The study struggles to connect its viral findings to practical restoration applications, and some key conclusions, particularly regarding methanogens, are based on limited data (only 1 virus and 2 hosts). There are also inconsistencies in the reporting of endemic viruses (54%) versus abundance changes (94%), and the temporal sampling/sampling details are limited.

Summary: This manuscript represents a strong contribution to viral ecology and ecosystem restoration but requires significant revision before publication. While the core findings are novel and important, particularly in linking viral dynamics to ecosystem health status, the manuscript needs stronger mechanistic evidence, better connection to broader ecological principles, additional functional validation, and improved theoretical framing. With these enhancements, particularly focusing on mechanistic understanding and practical restoration applications, the study could make a compelling contribution to the field.

We thank Reviewer #3 for the summary of our work's strengths and for highlighting important areas to improve. We have addressed each of the weaknesses raised:

- We have restructured our abstract and introduction to more clearly link soil viral communities to peatland restoration
- We have expanded the discussion of how our findings relate to peatland restoration in practice, including suggestions of how viral ecology knowledge could inform restoration monitoring and outcomes.
- We have qualified conclusions that relied on small sample numbers, such as the result involving a single methanogen-infecting virus, explaining the context and reliability of those observations.
- We have clarified the seeming inconsistency between “54% of viruses endemic to one status” and “94% showing abundance changes” by explaining that these statistics refer to different analyses (presence/absence across all sites vs. site-specific differential abundance).
- We have added details on sampling methodology (depth, timing, and site characteristics) as noted in responses to other reviewers.

Below, we provide detailed, comment-by-comment responses to Reviewer #3's specific points, with corresponding changes made to the manuscript.

Please see below more specific details:

SECTION SPECIFIC COMMENTS:

Abstract specific comments:

The abstract clearly presents the study's main findings, but the logical flow could be improved. For example, the abstract starts by stating viruses shape microbiomes and impact ecosystems, then jumps to peatlands degrading. While both points are important, it doesn't clearly establish why studying viruses specifically in peatlands is crucial. A stronger connection could be made between these opening statements. Additionally, while the study examines viral communities across different peatland health statuses, the findings don't clearly tie back to the initial problem of peatland degradation. While they found that viruses respond to ecosystem health and show specific patterns, the abstract doesn't explicitly explain how this knowledge helps with restoration efforts. The conclusion states that viruses are pivotal for restoration and climate mitigation, but the abstract doesn't fully support this claim. Making a clearer connection between the findings and their implications for restoration would strengthen the abstract.

We have now updated our abstract in our revised manuscript to address these concerns and to include our new findings.

Introduction specific comments:

The introduction effectively builds from broad context (peatlands' global importance) to specific focus (viral communities in restoration) but has several key areas that could be strengthened for greater impact. While it successfully establishes peatlands' significance for carbon storage and the threat of degradation, the connection between viral dynamics and restoration success needs more development. For example, the middle part talks about viruses and their role in ecosystems. It brings up some good examples from other peatland studies, showing how viruses affect carbon cycling and respond to water levels. But these examples feel a bit disconnected - like a list of facts rather than a story about why viruses matter for restoration. The biggest weakness is in explaining why we should care about viruses when trying to restore peatlands. The introduction tells us no one has studied viruses across healthy, damaged, and restored peatlands before. But it doesn't clearly tell us why these matters. What could we do differently in restoration if we understood viral patterns better? How might this knowledge help us restore peatlands more successfully? These questions aren't answered clearly enough. If space permits, I think it's also important to add some predictions to the end of the introduction. For example, what patterns would you expect to find or why those patterns would matter for restoration work? Adding these predictions would help readers understand why this research is useful, not just interesting. I don't think the paper had any predictions or hypothesis.

We appreciate the reviewer's thoughtful comments highlighting the need to strengthen the connection between viral dynamics and peatland restoration. In response, we revised the Introduction to better integrate the ecological significance of viruses into the context of peatland degradation and recovery. We restructured the section discussing prior studies to emphasize how viral community composition and function respond to

hydrological change and influence carbon cycling, using these examples to support a clearer rationale for their relevance to restoration. We now explicitly state that the functional reassembly of microbial communities during restoration likely involves viruses, and we highlight how changes in viral composition and function may reflect broader ecological recovery. Finally, we added three hypotheses at the end of the Introduction that reflect our expectations going into this study at lines 79-85:

“We hypothesized that (1) peatlands of varying levels of restoration would harbor unique viral assemblages, (2) viral populations would broadly mirror that of microbial hosts, with shifts in host metabolic functions across ecosystem states matched by their associated viral communities and their encoded functions, and (3) increased microbial activity in damaged sites would favor “piggyback-the-winner” dynamics, leading to elevated lysogeny and associations with fast-growing bacterial hosts.”

Results specific comments:

I have some questions about the metagenome assembly results in your study. You mention that co-assembling 66 samples yielded 22 high-quality metagenomes, which seems like a relatively low success rate of about 33% compared to other peatland studies. Could you clarify how much of this is related to particularly stringent quality criteria, wanting 90% completeness and less than 10% contamination?. Did you try different cutoffs to see if you could rescue more samples while keeping good quality? It would be helpful to understand whether this lower recovery rate was due to quality criteria, or technical challenges with specific sites, or other factors. I did find more information about this in the SI files, however it left me more confused. In the main text, you say co-assembling 66 samples yielded 22 high-quality metagenomes, but the SI shows these were actually co-assemblies that led to 935 medium and high-quality MAGs. Could you clarify what you mean by '22 high-quality metagenomes'? Are these the same as your 22 co-assemblies? Also, could you explain why you chose to make 22 co-assemblies from 66 samples - what was the logic behind this grouping? Did I miss anything? Or misunderstood your approach? Did you notice any patterns in which samples yielded high-quality assemblies - for instance, were natural sites more successful than disturbed ones?

We appreciate the reviewer’s careful attention and thoughtful comments. We recognize the confusion caused by our wording and would like to clarify our approach. We generated 66 individual sequencing libraries from replicate samples, specifically, triplicates for each of the 22 unique combinations of sampling site and ecosystem health status (3 replicates x 22 combinations = 66 libraries). These replicates were intentionally co-assembled into 22 separate metagenomes, with each co-assembly representing the combined reads from three replicate libraries from the same sampling condition. Thus, the reported 22 co-assemblies do not reflect a low success rate; rather, they represent our intended experimental design to leverage deeper sequencing coverage and enhance assembly quality, which co-assembly can offer for complex communities like soil (Riley et al., 2023 *Microbiol. Spectrum*; Delgado et al., 2022 *Microbiome*).

Regarding your question about the quality criteria: the “high-quality metagenomes” we referred to in the text were indeed the 22 co-assemblies themselves, from which we subsequently recovered 935 medium- and high-quality metagenome-assembled genomes (MAGs) by genome binning. The stringent completeness (>90%) and contamination (<10%) criteria you mention were applied during MAG binning, not to determine the number of co-assemblies. We have revised the manuscript text at lines 102-105 to clarify this distinction clearly and avoid further confusion:

“We sequenced the total community DNA of soil samples and co-assembled metagenomes by combining triplicate sequence read libraries from the same sampling site and ecosystem health status to yield 22 assemblies (one per site x ecosystem health status combination) that were overall of high quality (Supplementary Tables 1-2).”

Thank you for pointing out this opportunity to improve our explanation.

Additionally, could you discuss how this relatively low number of high-quality MAGs might have influenced your downstream analyses and conclusions about viral-host relationships?

We understand the reviewer’s concern about the apparent low number of high-quality MAGs. However, the recovery of 935 medium- and high-quality MAGs from 22 co-assembled soil metagenomes is actually robust and aligns well with, or surpasses, other recent soil metagenome studies (Trubl et al., 2021 *Microbiome*; Santos-Medellin et al., 2023 *Nat. Ecol. Evol.*; Huang et al., 2024 *Soil. Biol. Biochem.*). We acknowledge the reviewer’s point that incomplete genome recovery could potentially impact viral-host relationship analyses. We have clarified in the revised manuscript at lines 361-364:

“While some host-virus associations may have been missed due to incomplete MAG recovery and limitations due to in silico host prediction, the large number and diversity of MAGs obtained still provide a reliable foundation for understanding host-virus relationships in peatland soils.”

Although we certainly did not assemble and bin genomes for every microbe in the soil, we recovered enough of the community (including all dominant taxa) that our virus-host linkage analysis is representative. If anything, missing some MAGs (especially rare ones) means we might not have identified some virus hosts, but it would not falsely create host-virus links that don’t exist. So, the main potential influence is that our host diversity is a subset of the real diversity - we acknowledge that by avoiding any implication that our host list is exhaustive.

Can you provide more details on sampling? What depth were the samples collected from? Were they surface or deeper samples? How did you ensure valid comparisons across different sites? What was the water table depth, and how was it adjusted across sites? Also, when were the samples collected, and how might seasonality affect the results? Depth could have a strong impact on the findings.

We agree that more details on the sampling methodology were necessary. We have now included additional sampling information in the methods section of our revised manuscript at lines 430-438. Additionally, we added a citation to a companion study that contains greater detail on the sampling methodology and site characteristics (Pallier et al., 2025 bioRxiv). The cited manuscript states that all sites were sampled in the warmer, drier months (between May and October 2021), thus the impacts of seasonality on our results are minimal. All samples came from the top 10 cm depth.

I am also a bit confused about how you compared their viruses to existing databases. You say you found more matches with other databases than within your own study, but it's not really clear what this means? It would really help to know how these patterns compare to other types of soil. Are peatland viruses more widespread than viruses in, say, forest soils? Also, are certain types of viruses better at spreading around than others? Can you expand a little bit more on Figure 2 in the text.

We appreciate the reviewer's question regarding comparisons with existing databases. Our intention in comparing our viruses to published soil virus databases was primarily to assess novelty and the overall level of representation of peatland viruses in existing collections. We have now added the following in the main text at lines 161-168 to reflect this intention:

"We also assessed whether the viral genomes we identified were largely novel or instead represented in previously published soil virus databases... many viral genomes clustered with known soil viral genomes from other ecosystems, indicating they are not wholly unique to peatlands at the genus level."

We aimed to demonstrate that peatland viruses identified in our study are not entirely novel but rather share significant similarities at the genus level with viruses from various soils globally. Expanding this comparison to include detailed analyses against forest soils or other soil types was beyond the scope of this particular study.

Lines 115-134 Earlier the authors said 54% of viruses were 'endemic' to specific ecosystem states, but here they are showing that 94% of viruses change in abundance across states. This seems contradictory - are these viruses present but rare in some states and abundant in others?

We recognize that our initial wording may have caused confusion regarding the distinction between the 54% endemicity statistic and the 94% differential abundance statistic. The 54% specifically refers to the presence or absence of viral species clusters across different ecosystem health statuses (i.e., viruses present exclusively in one ecosystem health status and absent in the others). In contrast, the 94% statistic pertains to differential abundance testing, which was performed independently at each site. This statistic considers virus species clusters that exhibit significant changes in their relative abundance across ecosystem health statuses within individual sites.

To illustrate this, consider a virus genome assigned to a species cluster that is present in only one health status across all sites (thus, included among the 54% "endemic" viruses). Because this virus genome is present in one health status but absent from the other two, it inherently exhibits a significant difference in abundance across ecosystem health statuses, a difference that would be detected as significant by DESeq2. Therefore, such a virus genome is simultaneously endemic to a single health status and also classified as differentially abundant across statuses.

Now, consider a different virus genome assigned to another species cluster, one that is detected in two or even all three ecosystem health statuses according to our detection criteria. This genome is not counted among the endemic viruses that collectively constitute the 54% statistic. Nevertheless, even though this virus genome occurs in multiple ecosystem health statuses, its abundance may still be significantly higher in one status compared to the others, which DESeq2 is designed to identify. Consequently, this virus genome contributes to the 94% statistic for viruses that were found to differ significantly in relative abundance across ecosystem health statuses, despite not being classified as endemic.

Furthermore, as stated in the supplementary text, the same virus species can exhibit differential enrichment patterns at different sites, owing to the fact that the species-representative genomes are representatives for clusters of virus genomes dereplicated at 95% identity (see methods), contributing to the higher percentage of viruses identified as differentially abundant.

We have revised the manuscript at lines 86-94 of the Supplementary Text to clearly articulate these distinctions and ensure there is no ambiguity:

“Note that endemicity (presence exclusively in one health status across all sites), as determined by our presence/absence analysis (see main text), and differential abundance (significant abundance variation between statuses within individual sites) represent complementary but distinct measures of viral distribution. Viruses endemic to a single status inherently differ in abundance from statuses where they are absent, while viruses present in multiple statuses can still show significant abundance shifts, contributing to the higher proportion (94%) of differentially abundant viruses reported. Additionally, virus species representatives are clusters dereplicated at 95% identity, allowing for variable differential abundance patterns across sites.”

We direct the reader to this section at line 180 of the Main Text in case they need further clarification.

lines 136-153. Could you provide more context about the virus-host abundance discordance - what mechanisms might explain cases where viral populations increase while host populations remain stable?

This discordance, where viral abundances increase independently of detectable changes in host abundance, can occur due to several ecological mechanisms. For

instance, increased viral abundances could reflect enhanced viral production or lytic activity triggered by environmental stressors such as changes in soil chemistry, nutrient availability, or alterations in host metabolism. Under such conditions, host cells might become more susceptible to viral infection or shift from lysogenic to lytic infection cycles, promoting viral replication without affecting overall host numbers significantly. Additionally, certain viral populations might have broader host ranges, enabling them to proliferate by infecting alternative host species if their primary hosts remain stable in abundance. Furthermore, host populations experiencing environmental stress may undergo physiological or genetic changes, altering their susceptibility or defense mechanisms, thereby increasing viral replication success.

We agree this warrants clarification and we have explicitly discussed these potential mechanisms in the revised manuscript at lines 382-387.

lines 154-187 The methanogen virus conclusions are based on very small numbers (1 virus, 2 hosts) - how reliable are these patterns? Could you provide power analyses or additional validation? Also, for the reported changes in fermentation (-22%) and carbohydrate degradation (-15%), were these changes statistically significant?

We agree that conclusions drawn from a very small number of observations must be handled carefully. In the original text, we noted that a virus infecting methanogen hosts was enriched in restored soils. We have now expanded and qualified this result to reflect the limitation of $n = 1$ virus (and $n = 2$ host MAGs). To focus on other interesting trends justified by larger sample sizes, we now highlight our observed trends in viruses that infect microbes with the ability to perform oxidative phosphorylation ($n = 451$, lines 218-223). We have thus moved the result for the viruses of methanogens to the Supplementary Text (lines 109-113) and state:

“Although this observation involves only a single viral species representative genome and two host representative genomes, each represents an aggregated species-level population across multiple samples and replicates. Thus, the $n = 1$ virus or $n = 2$ hosts here should not be taken as literally a single individual, but rather as capturing an entire lineage present in the community.”

In other words, the “1 virus” is a species-representative genome that was present in several samples (in this case, the restored peat replicates). We have added this clarification to assure the reader that the finding, while based on a small number of lineages, is still biologically meaningful.

Regarding the percentage changes in fermentation and carbohydrate degradation functions: these were aggregate relative abundance changes (for trend groups) and were not subjected to statistical tests in the original analysis because they are derived from normalized ratio data without per-replicate variance (each value was from an aggregated ratio of ratios). We realize this could be confusing, so we have now explicitly noted in the Methods/Results that these percentage changes are descriptive. In the Methods (Lines 728-732), we have now added:

“The associated reported changes in relative abundance ratios reflect aggregated populations at the species level rather than individual replicates, limiting the applicability of traditional statistical significance tests. These observations highlight biologically meaningful trends rather than statistically validated differences.”

Furthermore, when presenting these changes, we avoid implying statistical significance and use phrasing like “showed a decrease in damaged soils (-22%).” Also, we have now added the following to clarify this in lines 238-240 of the results and direct the reader to the statement cited above:

“It is important to note that these percentage changes reflect descriptive trends based on aggregated ratios and were not subjected to null hypothesis testing (see Methods).”

Additionally, while you show viral protein adaptation across ecosystem states, how might these adaptations influence restoration success?

While exploring how the inferred viral protein adaptations might directly influence restoration success is intriguing, we feel that expanding on this aspect would risk increasing the manuscript length and extending beyond our current scope. Therefore, we have chosen to maintain our existing discussion of these protein adaptations as is.

lines 222-256) The finding of only 13% lysogenic viruses seems potentially low - how does this compare to other soil environments? Could technical limitations in identifying lysogenic viruses affect this percentage? You show opposing trends in absolute versus relative abundance of lysogenic viruses in damaged soils - what are the implications for restoration practices?

We acknowledge that the identified 13% proportion of lysogenic viruses in our dataset may initially appear low. However, this figure reflects only the subset of viral genomes containing confidently annotated lysogeny-associated genes (e.g., integrases and excisionases). Predicting viral lifestyles from metagenomic sequences alone is challenging due to the incomplete annotation of viral genomes and limitations in identifying lysogeny markers (Shang et al., 2022 *Brief. In Bioinform.*). Therefore, this percentage (13%) does not represent the total proportion of lysogenic viruses in peatland soils, but rather the minimum confirmed fraction. Similar percentages have been reported in other soil metagenome studies due to these technical limitations (Muscat et al., 2023 *Microbial Ecology*; Tang et al., 2023 *ISME J.*). We have addressed this concern by expanding our discussion of technical limitations (as also noted in response to Reviewer 2’s main issue, please see our response above).

The reviewer correctly notes that absolute abundance trends differed from those normalized by total viral populations. We agree that this was unexpected. Upon further inspection of sample distributions and diagnostic statistical analyses, we now present lysogenic virus abundance as both raw and normalized sample-wide means, shown in the new Extended Data Figure 5 and the revised Figure 5B. Although the magnitude

and statistical significance of comparisons differ slightly between the raw (Extended Data Figure 5A) and normalized data (Extended Data Figure 5B, Figure 5B), the overall trends remain consistent: lysogenic virus abundance increases as ecosystem health declines. This observation is further confirmed by modeling lysogenic virus abundance as a function of our newly introduced continuous ecosystem health index (Figure 5C). We have revised this section with our new results in lines 311-315.

We agree that this warrants clearer discussion in its relation to restoration, and we have included the following wording in lines 383-403:

“... these viral dynamics are sensitive to changes in environmental disturbances, such as altered nutrient availability, redox changes, or changes in soil chemistry following degradation, which may trigger switches between active lytic and lysogenic replication... Ultimately, shifts between lysogenic and lytic viral replication strategies could serve as indicators of population densities of hosts with specific ecosystem functions in peatlands undergoing restoration.”

Could you explain how you selected the eight key metabolic functions for analysis, and whether other important functions might have been missed.

We specifically selected these eight metabolic functions (oxidative phosphorylation, methanogenesis, fermentation, carbohydrate degradation, aromatics degradation, assimilatory sulfate reduction, dissimilatory sulfate reduction, and thiosulfate oxidation) due to their recognized ecological importance and direct relevance to carbon and sulfur cycling processes in peatland soil ecosystems (Brooks Avery et al., 2003 Biogeochemistry; Pester et al., 2012 Front. Microbiol.; Dalcin Martins et al., 2017 Global Change Biol.; Robinson et al., 2023 Mires and Peat; McGivern et al., 2024 Nat. Microbiol.; Richy et al. 2024 Soil. Biol. Biochem.). We further stress this in the revised manuscript in the methods in lines 671-676:

“We focused on eight metabolic functional categories relevant to peatland soil ecosystems—oxidative phosphorylation, methanogenesis, fermentation, carbohydrate degradation, aromatics degradation, assimilatory sulfate reduction, dissimilatory sulfate reduction, and thiosulfate oxidation—due to their recognized ecological importance in soil carbon and sulfur cycling and their expected sensitivity to peatland disturbance and restoration (Brooks Avery et al., 2003 Biogeochemistry; Pester et al., 2012 Front. Microbiol.; Dalcin Martins et al., 2017 Global Change Biol.; Robinson et al., 2023 Mires and Peat; McGivern et al., 2024 Nat. Microbiol.; Richy et al. 2024 Soil. Biol. Biochem.).”

While other metabolic functions undoubtedly occur in peatland microbiomes, our focus was explicitly on processes with established significance to peatland carbon and sulfur cycling dynamics. We acknowledge the potential relevance of additional functions and encourage future research to explore these other metabolic pathways.

Discussion section:

While the results section provides a lot insights and potentially new findings, unfortunately the discussion falls short in several important ways. While the authors

found interesting patterns in viral communities across peatland sites, they don't clearly explain how these findings could help restore peatlands. This is my major critique of the paper. The authors mentioned multiple times that viruses are 'pivotal' and 'critical' for restoration, but never tell us what this means in practice. Another issue is that the authors talked about viruses controlling methanogens -I found this very exciting with a lot of implications however this big claim is based on just one virus and two host genomes. How important is this virus then? (maybe expand on this on LINES 284-287.) The authors also skip over some big questions from their results. They never explain why they found different numbers for endemic viruses (54%) versus viruses that change in abundance (94%).

We thank the reviewer for highlighting the need to clarify the practical implications of our results for peatland restoration. Previously, our manuscript lacked explicit connections between viral community findings and restoration practices, primarily due to limited environmental data. This has now been addressed by providing more analysis of the associated environmental data and an ecosystem health index (see our response to reviewer #1, above). We agree that conclusions drawn from a small number of lineages must be approached cautiously. We have addressed this concern in detail (see our comments above), clarifying the representative nature of the virus and host genomes analyzed. We hope these revisions fully address your concern.

We appreciate your pointing out the apparent discrepancy between endemic viruses (54%) and differentially abundant viruses (94%). This important point has been carefully addressed above, with new additions to the discussion section, where we explained the distinct methodologies used for presence/absence (endemicity) and differential abundance testing at individual sites.

It will be good if the authors expand in the main text or in supplementary on how the methods of choice, and while might undercount viruses, how this could affect their conclusions. You only looked at 22 MAGs, how representative is that of real dynamics?> were these 22 MAGs representative of the full community?

Please see our response above. This has been clarified in detail.

The ending feels weak. Instead of offering specific next steps, the authors give vague suggestions like 'future research should explore mechanisms'. They don't describe any experiments that could help prove their ideas about viruses in restoration. Most importantly, they don't explain how restoration experts could use these findings to improve their work. For a paper focused on peatland restoration, this is a significant gap. If the goal of the paper is simply to characterize viruses, I think that's fine. However, the authors will need to remove the claim that this will help with restoration, as it doesn't follow from the current findings.

The broader ecological context could also be better developed to demonstrate impact beyond peatland systems. The ecological context could be enhanced by better connecting findings to general ecological theory and expanding implications beyond

peatland systems. The presentation could also be improved by emphasizing novelty more clearly in the introduction and strengthening connections to broader ecological principles and climate change implications

We appreciate this suggestion highlighting the need for more specific recommendations and future research directions at the conclusion of our manuscript. We have revised our discussion to explicitly include concrete next steps and potential experimental approaches that would complement and validate our findings, thereby offering clearer practical value to peatland restoration practitioners. Specifically, we suggest studies to verify key mechanisms identified here (e.g., lysogenic to lytic switching) and explicitly link viral activity to measurable peatland biogeochemical outcomes relevant to restoration.

Moreover, we have now incorporated explicit recommendations on how restoration practitioners might practically apply our findings to enhance peatland restoration outcomes (lines 406-417, 423-427). We suggest incorporating the viral and host community metrics we present into restoration monitoring and design. These metrics can be related to our ecosystem health indices to assess restoration success. We hope these revisions address the reviewer's concerns and substantially strengthen both the clarity of the practical restoration implications and the manuscript's broader ecological relevance.

One minor thing: I couldn't access this link: <https://doi.org/10.6084/m9.figshare.28143446>. (DOI Not Found). I am assuming it is not public yet

Reviewer #3 (Remarks on code availability):
the DOI link was not working. The GitHub link worked but I didn't check the code.

Please see our response to reviewer #1 comment 7, above.

Reviewer #1 (Remarks to the Author):

Thank you to the authors for considering and addressing my comments. I appreciate the inclusion of the environmental data and the health index, and feel they provided useful context for the site categories. All my concerns have been addressed.

We are glad that all your concerns have been addressed. Thank you for your positive assessment.

Reviewer #1 (Remarks on code availability):

The github looks comprehensive. I understand the reviewers are keeping the data private before it is public, but the github links to Bioproject and figshare links will need to be made public.

The NCBI BioProject PRJNA1203648 is now public along with all associated sequence data. The FigShare link to the additional MAGs and vMAGs will be made public upon publication of the manuscript automatically through the Nature Microbiology manuscript tracking system.

Reviewer #2 (Remarks to the Author):

No further comments. Thank you for addressing my previous questions and requests.

We are happy that we've addressed your questions and requests sufficiently. Thank you for your comments.

Reviewer #2 (Remarks on code availability):

No further comments.

Reviewer #3 (Remarks to the Author):

I appreciate that the authors responded thoroughly to my comments and the other reviewers' feedback and made substantial improvements to the manuscript. I especially liked the new environmental data and the ecosystem health index. It makes a lot stronger case for their health categories. Also, thanks for clearing up the confusion between the 54% endemic viruses versus 94% differentially abundant viruses, and the MAGs versus co-assemblies.

We appreciate your thoughtful and positive feedback, and thank you for recognizing the improvements made throughout the revision process.